# Electric vehicle battery chemistry affects supply chain disruption vulnerabilities

Anthony L. Cheng[1], Erica R. H. Fuchs[1], Valerie J. Karplus[1,2] & Jeremy J. Michalek [1,3,4] ✉

We examine the relationship between electric vehicle battery chemistry and supply chain disruption vulnerability for four critical minerals: lithium, cobalt, nickel, and manganese. We compare the nickel manganese cobalt (NMC) and lithium iron phosphate (LFP) cathode chemistries by (1) mapping the supply chains for these four materials, (2) calculating a vulnerability index for each cathode chemistry for various focal countries and (3) using network flow optimization to bound uncertainties. World supply is currently vulnerable to disruptions in China for both chemistries: 80% [71% to 100%] of NMC cathodes and 92% [90% to 93%] of LFP cathodes include minerals that pass through China. NMC has additional risks due to concentrations of nickel, cobalt, and manganese in other countries. The combined vulnerability of multiple supply chain stages is substantially larger than at individual steps alone. Our results suggest that reducing risk requires addressing vulnerabilities across the entire battery supply chain.

Vehicle electrification is an important pathway for reducing transportation's greenhouse gas emissions. Transportation end use emissions (across all types of transportation) accounted for 27% of emissions in the United States[1] and 24% of worldwide emissions in 2020[2]. Automaker battery design choices, which very often involve specific choices of battery chemistries and firm boundaries (as seen in Supplementary Text S1-1), imply reliance on particular critical materials and associated processing infrastructures. The rapid scale-up of electric vehicle (EV) battery production implies a massive increase in demand for these critical minerals, with projected increases of 5 to 40 times 2020 demand in 2040, depending on the material[3–5].

To the extent that battery supply chains rely on one or a few countries for specific process steps, many governments have grown concerned about vulnerability to supply disruptions and some have enacted policies to incentivize domestic production. Recent or announced legislation in the US[6,7], Europe[8], and elsewhere[9] requires minimum percentages of domestic manufacturing or sourcing from allies (such as free-trade agreement countries in the US context) to qualify for incentives or avoid fines or to meet regulations on carbon footprint and responsible sourcing. As a result, measurements of whole-supply chain vulnerabilities are required to better help understand economic, sustainability, and national security risks. Such measurements may help policymakers evaluate incentives and promulgate regulations on the basis of specific supply chain interventions, such as to reduce such risks, promote domestic production, or internalize economic benefits.

Today, electric vehicle batteries mainly use lithium ion chemistries[3,5]. The primary lithium-ion cathode chemistries are NCA (lithium nickel cobalt aluminum oxide), NMC (lithium nickel manganese cobalt oxide), and LFP (lithium iron phosphate), which depend on varying amounts of four primary critical minerals: lithium, nickel, cobalt, and manganese, as identified in studies of mineral criticality for battery cathode materials[10–12]. Discussion of other minerals that may be deemed as critical or could become critical, including phosphorus, aluminum, and iron, as well as the criticality of other battery chemistries under development, is provided in Supplementary Text S1-2.

The battery supply chain can be separated into three segments: upstream (mining and extraction of raw materials), midstream (processing of raw materials into battery-grade components), and downstream (cell and pack manufacturing, as well as end-of-life recycling

[1]Department of Engineering and Public Policy, Carnegie Mellon University, Pittsburgh, PA, USA. [2]Wilton E. Scott Institute for Energy Innovation, Carnegie Mellon University, Pittsburgh, PA, USA. [3]Department of Mechanical Engineering, Carnegie Mellon University, Pittsburgh, PA, USA. [4]Department of Civil and Environmental Engineering, Carnegie Mellon University, Pittsburgh, PA, USA. ✉e-mail: jmichalek@cmu.edu

and reuse)[13]. The supply chains for the critical minerals in these batteries differ in terms of the geography of raw material production (Fig. 1), although a few countries produce the majority of supply for each critical mineral. Arguably the most important choice is the selection of cathode material, as cathodes are over half of the cost of a battery cell and largely determine crucial battery characteristics such as energy density and charging speed[14]. While other components such as the anode (graphite)[15] and electrolyte (typically lithium salt solutions)[16] may also suffer from vulnerabilities in their supply chain, the choices among these components are far more limited and thus do not offer the same kind of options to reduce vulnerability by changing technology choices. Furthermore, production data were also much more limited and thus these battery components were excluded from our scope. Most automotive manufacturers around the world are exercising increasing levels of control over their material supply chains as they design batteries for their vehicles (see Supplementary Table 1). As such, this study focuses on the decision-making of firms in the upstream and midstream supply chains, ending at the current typical firm decision boundary, the choice of battery cathode chemistry.

While upstream mining is geographically distributed by material, China dominates every part of the midstream and downstream supply chains of all materials, from material processing and refining to electric vehicle production. What is unclear from these production data is the degree to which electric vehicle battery material supply chains have vulnerabilities to specific countries when taking into consideration all supply chain steps. About 74% of mined lithium[17] and 57% of mined cobalt[18] is used in lithium ion batteries, but only a portion of lithium ion batteries are used in electric vehicles. Furthermore, only about 11% of nickel[19] and 2% of manganese[20] is used to make batteries of any kind.

Several streams of literature have developed vulnerability-related concepts and metrics, such as literature on energy security, materials criticality, material flow analysis, input-output analysis, and supply chain disruption propagation. While a detailed literature review is provided (Supplementary Text S2), we highlight below a few key studies that have framed discussion of battery material supply chains, country-based risks, and measures of vulnerability below.

A number of studies in the material flow analysis literature examine the structure of these supply chains in detail. Using a combination of trade data and production data at each step of the supply chain, these studies describe the relationships between geographies of production, with countries as the unit of geographic resolution. In the context of battery materials, parts of this literature focus on specific stages of the value chain, e.g. raw materials and mining, while others encompass all steps, but the scope is almost always global and limited to one specific battery material – lithium[21], cobalt[22,23], nickel[24], manganese[25]. We use these methods to reveal the underlying material

production and trade networks that undergird the supply chain flows in our analysis. By including the cathode material manufacturing step, we can then compare the relative demand for each material that is required to fulfill material demand for batteries, a novel addition not found in studies of individual critical mineral material flow analyses.

Several materials criticality studies have included measurements of country-based production concentration in assessments of vulnerability or risk[10,26,27]. While these studies combine measurements of supply chain structures in ways that aggregate across multiple materials, these studies tend to focus on the relative importance of critical minerals. However, in trying to combine several metrics, these studies can obfuscate the vulnerability of risk caused by specific countries. On the other hand, studies in the recently developing supply chain disruption propagation literature[28–31] identify specific countries, particularly China, as critical nodes in battery material trade networks, by calculating the avalanche size of a disruption – the number of countries affected by the removal of supply or trade from one country. However, these studies do not combine multiple critical minerals and also make the general assumption that disruptions occur as proportional losses of trade from the disrupted country, which then propagate to other countries, which does not encompass the full scope of realities of how disruptions occur.

Furthermore, across all of these studies, the treatment of uncertainty varies substantially. For example, the material flow analysis literature provides insight into methods for accounting for missing trade, but these uncertainties are not always incorporated into materials criticality measurements. A recent study[32] evaluated the feasibility of meeting the recently passed U.S. Inflation Reduction Act's goals for minimum critical mineral requirements across all battery materials for the United States while accounting for uncertainty in imports and maximum material availability. They found that achieving market-value-based targets may be possible with NCA batteries but not necessarily for LFP or NMC batteries. However, this study does not consider additional supply chain steps beyond direct imports to the United States, which can be uncertain.

In this work, we suggest there is an important gap in the existing literature given a lack of analysis of the vulnerabilities in the global flow of multiple battery materials between countries as sources of both supply and demand across stages of the supply chain, particularly in terms of physical quantities of materials. While the criticality literature typically takes the perspective of an importing focal country, our study instead assesses global supply chain dependence on exporting focal countries. Moreover, existing studies do not explore multiple sources of uncertainty. While we do not attempt to determine potential causes of risks and vulnerabilities, nor assess the probability of such risks or vulnerabilities occurring, we aim to provide a quantitative metric to

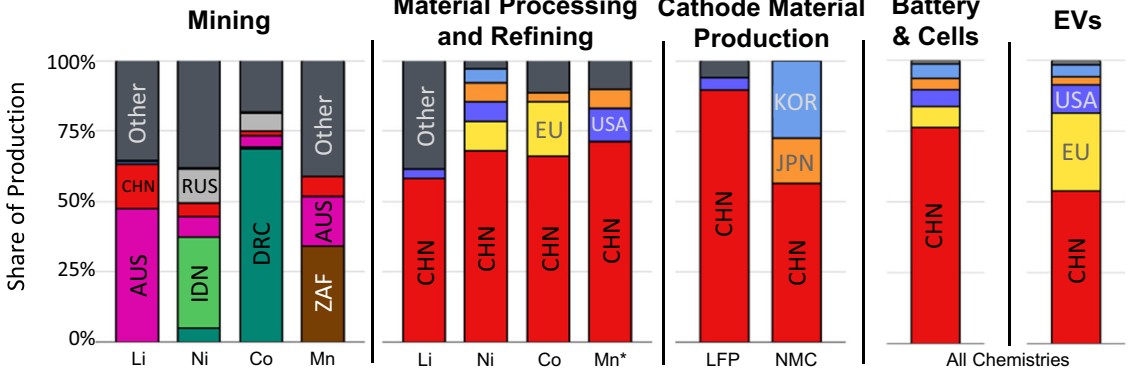

**Fig. 1 | Geographical Distribution of EV Battery and Material Supply Chains[5,17,59].** Li lithium, Ni nickel, Co cobalt, Mn manganese (*electrolytic manganese dioxide represents refined manganese). The following codes are used to represent countries or regions: CHN China, AUS Australia, IDN Indonesia, RUS Russia, DRC Democratic Republic of the Congo, ZAF South Africa, KOR Republic of Korea, JPN Japan, USA United States of America, Other any other country not explicitly listed here.

help assess vulnerabilities and evaluate potential actions to reduce vulnerabilities. In addition to specific insights about the supply chains for electric vehicle batteries, including the explicit inclusion of the cathode material manufacturing step, we provide a novel methodology incorporating material flow analysis methods with optimization methods to account for the impact of uncertainty on measurements of vulnerability.

## Results

We examine the impact of electric vehicle battery design choices on the vulnerability of battery material supply chains. We first describe a static distribution of the supply chains, accounting for trade and production of all materials related to the four primary critical battery minerals used in battery cathodes (lithium, cobalt, nickel, and manganese). We then measure a vulnerability index for a country (or combination of countries) as the total percentage of the end product (in this case, battery cathode material) produced with materials that are either manufactured in or traded from that country, at any step of the supply chain, while accounting for uncertainty due to sourcing of materials at each step and to missing trade data. Our analysis focuses on LFP (lithium iron phosphate) and NMC (lithium nickel manganese cobalt) as they have the minimum (1) and maximum (4) number of critical minerals covered in this assessment.

### Battery cathode material flow supply chain mapping

We describe the global supply chains for lithium in Fig. 2 and for cobalt, nickel and manganese in Fig. 3., considering the known demand for various lithium ion battery cathode materials. From left to right, each figure shows the portion of global material supply in each country for raw material mining and extraction, raw material trade, refining and processing, refined material trade, and cathode production, which includes trade of scrap or recovered materials that may be involved in the battery material supply chain. For each section labeled as trade (i.e. flows within each rectangular boundary), the flows are between countries, and sections labeled as processing (i.e. flows crossing the rectangular boundaries) are wholly within the country. Boxes that are bisected by the rectangular boundaries could be involved in either trade or production, but are uncertain. We note that not all mined or refined materials are used for battery cathodes. These diagrams map the countries and trade flows that are explicitly involved in these battery material supply chains, as described by the literature. Details about the methodology and data underlying in Figs. 2 and 3 are found

in the Methodology section and Supplementary Text S3, with the simplified version of the supply chain modeled in this paper presented in Supplementary Text S1-1.

Figure 2 shows that most lithium used in battery production in 2020 was extracted in Australia (49%), Chile (27%), China (16%), Argentina (7%), and the US (1%), where values are rounded to the nearest percentage point. These countries generally processed the lithium they extracted, with the exception of Australia – 99% of which was shipped to China and the remaining 1% of which was shipped to the US. The primary reason for this is because Australia and China are the main producers of spodumene rock, which is refined to lithium products, while the other countries produce brines that must be refined into lithium products in situ[21,28]. About 31% of the raw Li acquired by China and 3% of Li from Chile appears to have been used or refined in non-battery contexts. The remainder was processed into battery-grade lithium by China (59%), Chile (29%), Argentina (9%), and the US (3%). (Chilean imports and exports exceeded the reported production, which is why Chile had both raw materials not accounted for in battery-related refining and missing additional refining production.) Refined lithium was then traded to China (55%), South Korea (16%), Japan (12%), the US (5%), Canada (1%) and other countries that do not produce cathode materials (TNPC, 12%). Not all of the lithium is used in battery cathode production – 41% (China), 44% (South Korea), 29% (Japan), 95% (United States), and 55% (Canada) of each country's production were used in non-battery cathode related products, respectively. Finally, LFP cathode material was produced with refined lithium from China (90%), the US (4%) and Canada (6%), while NMC cathode material is produced using refined Li from China (57%), South Korea (27%), and Japan (16%).

The other Sankey diagrams in Fig. 3 can be interpreted similarly. In general, we see that a large amount of cobalt, nickel, and manganese is used for products other than batteries. It is particularly notable that a large amount of nickel and manganese are transformed into products that are not considered battery related, so even though large producers such as Indonesia, Australia, and Gabon exist, the amount of influence they have on battery material supply chains may be limited. It is also important to note that the amount of uncertainty present in these figures is relatively high, particularly for Cobalt and Nickel, as there is a significant amount of uncertainty in supply in both the raw and refined material steps. While we capture the effect of this uncertainty in our vulnerability index calculation, we note that gathering better data along the supply chain is important to continue to refine and inform our understanding of vulnerability and risk.

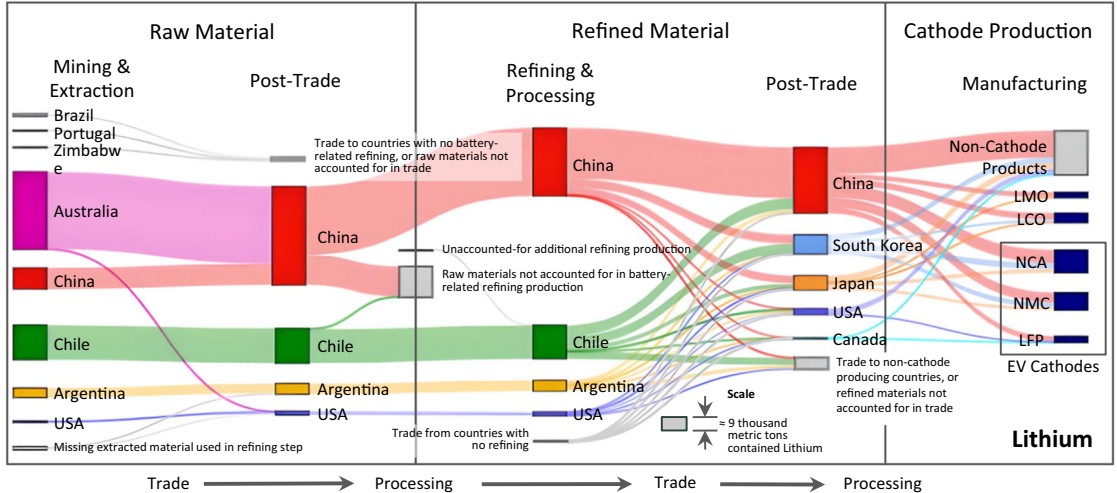

**Fig. 2 | A Sankey diagram for global flows of lithium that available data suggest are involved in battery material supply chains.** See Supplementary Text S3-2 for further details and data sources. Note that the scale of production – 82,500 metric tons of lithium were mined and extracted in 2020 (excluding missing production). Note that LMO (lithium manganese oxide) and LCO (lithium cobalt oxide) cathodes are currently rarely used in electric vehicle batteries.

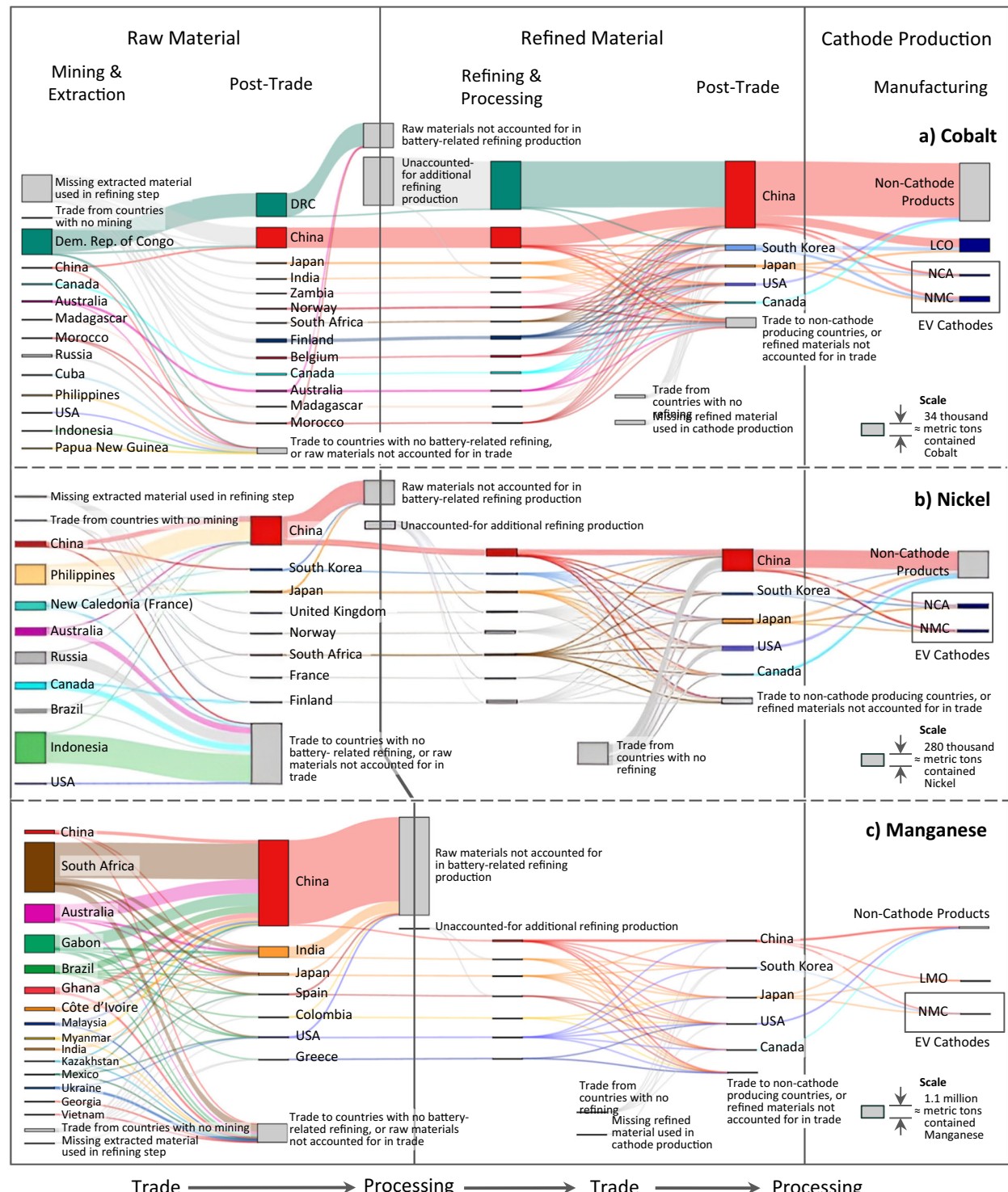

**Fig. 3 | Sankey diagrams for global flows of cobalt, nickel, and manganese that available data suggest are involved in battery material supply chains.** See Supplementary Text S3-2 for further details and data sources. Note that the scale of production differs – 142,000 metric tons of cobalt (**a**) were mined and extracted in 2020 (excluding missing data), compared to 2.5 million metric tons of nickel (**b**) and 5.67 million metric tons of manganese (**c**). Note that LMO (lithium manganese oxide) and LCO (lithium cobalt oxide) cathodes are currently rarely used in electric vehicle batteries.

## Measuring a vulnerability index

Based on these flows, we can quantify measures of supply disruption vulnerability for a focal country or focal set of countries. In this analysis, we define an index of vulnerability to supply disruption from a country $i$ as the total percentage of the end product (in this case, battery cathode material) produced with materials that are either manufactured in or traded from country $i$ at any step of the supply chain. We note that uncertainty in this measurement exists due to:

1. unobserved intranational material flows: the source data report aggregate production within each country and aggregate trade between pairs of countries without observing which materials produced or acquired by a country follow which production and export paths.

**Table 1 | Approaches to bounding uncertainty of intranational and international material trade flows, for assessing the portion of materials that are processed by or exported from a particular focal country**

| Case | Unobserved intranational material flows | Unobserved international trade and indirect trade |
|---|---|---|
| **Base Case:** Proportional flows | Each country's inputs assigned *proportionally* to its outputs | Also proportionally distributed |
| **Optimistic Case:** Minimum portion involving focal country | Each country's inputs are assigned to its outputs using network flow optimization to *minimize* the portion of materials used for the focal battery chemistry that include materials processed or exported by the focal country. | Indirect trade and unobserved trade not assigned to focal country |
| **Pessimistic Case A:** Maximum portion involving focal country, observed direct trade only | Each country's inputs are assigned to its outputs using network flow optimization to *maximize* the portion of materials used for the focal battery chemistry that include materials processed or exported by the focal country. | Indirect trade and unobserved trade not assigned to focal country |
| **Pessimistic Case B:** Maximum portion involving focal country, all (observed and unobserved) trade | Each country's inputs are assigned to its outputs using network flow optimization to *maximize* the portion of materials used for the focal battery chemistry that include materials processed or exported by the focal country. | Indirect trade and unobserved trade assumed to originate from the focal country |

2. unobserved international trade: data that should exist to satisfy mass balances when producing battery materials but is not present in the trade data.
3. indirect trade: trade from countries without known processing capability, such as intermediary-based trade used to avoid tariffs[33].

Table 1 summarizes how we manage these three sources of uncertainty, defining a base case using proportionality assumptions common in the literature and bounding uncertainty with optimistic and pessimistic cases. These uncertainties also only apply in the context of trade data, as we assume the production data to be accurate.

Our vulnerability index aims to estimate the potential for a disruption in one country (or set of countries) to affect the overall supply of battery cathodes given current supply chain flows. As battery production grows and supply chains shift, vulnerability indices can be updated to reflect changing interdependencies. Our vulnerability index does not capture all possible factors that affect critical material vulnerabilities. For example, concentration of raw material reserves countries may affect the long term interdependencies in ways that are not reflected by current material flows[34,35].

As an illustrative example, we present an analysis of the global disruption vulnerability index of LFP and NMC cathode production for China, the country with the most extensive involvement in battery material supply chains (as described in Fig. 1). This analysis is performed for other countries and regions in Supplementary Fig. 7 and Supplementary Fig. 8. Summaries for each case are presented in Fig. 4, with the contribution of each step of the supply chain to the vulnerability index shown as a proportion of the end cathode material affected.

The most simple method to measure a vulnerability index, in line with ideas from the supply disruption propagation and input-output literature, is to calculate a point estimate proportional case. To do this, we only take into account the relative proportion share of imports and domestic production for determining the relationship of each country with the upstream step. Results for the proportional LFP and NMC analyses are summarized in Fig. 4a, e. For LFP, 90% of cathode production takes place in China, and the US and Canada make up the remaining 10%, suggesting a vulnerability index of at least 90%.

We bound uncertainty in the vulnerability index in Fig. 4b, c, f, g using network flow optimization to minimize or maximize the portion of the supply chain that could flow through China, given uncertainty about which imports for each country map to which paths. Using only the trade data that is present and accounting for the first type of uncertainty (unobserved intranational flows), we estimate minimum and maximum bounds, as for example we do not necessarily know where the observed production of LFP in China sources lithium from.

This corresponds to corresponding to an optimistic case and pessimistic case A. With our base case proportional flow estimates [optimistic estimate to pessimistic case A estimate], we estimate that 17% [0% to 33%] of refined Li imports to the US and Canada are refined in China, increasing the vulnerability index by +2 [+0 to +3] percentage points to a total of 92% [90% to 93%] of cathode material production involving China, depending on which US and Canada imports are used for LFP cathode production. However, it is important to recognize that this scenario, while pessimistic, is not a guaranteed upper bound, as we can have uncertainty due to non-observed and indirect trade that could be attributed to production in China (uncertainties 2 and 3). Subsequently, we include a second, more extreme upper bound (Pessimistic Case B) by simultaneously assuming that all indirect trade and missing materials originate from China, resulting in the vulnerability index for LFP increasing by +7 percentage points between the two pessimistic cases to 100%, as seen in Fig. 4d. These results suggest that LFP battery material supply chains are highly vulnerable to a disruption in China, as even in the most optimistic case the vulnerability index is over 90%. While not noted on the figure, further analysis indicates that even if all cathode production were moved out of China, an estimated 71% of LFP cathode material would use Li inputs produced or traded from China in the proportional case.

For NMC, 57% of cathode production takes place in China, and South Korea and Japan make up the remaining 43%, suggesting a vulnerability index of at least 57%. We estimate that 50% [0% to 100%] of refined Li imports to South Korea and Japan are refined in China, increasing the vulnerability index by +21 [+0 to +43] percentage points to a total of 78% [57% to 100%] of cathode material production involving lithium and China, depending on which South Korean and Japanese imports are used for NMC cathode production. Assigning indirect trade and missing data to China does not change this range (i.e. in this case, the missing and indirect trade has no effect), because the index cannot increase beyond 100%. Furthermore, though it is similarly not noted on the figure, we note that even if all cathode production were moved out of China, an estimated 68% of NMC cathode material would use Li inputs produced or traded from China in the proportional case. Example solutions for the other NMC materials, similar to Fig. 4, are in Supplementary Text S4-1.

Using this method, we also calculate this disruption vulnerability index for the NMC supply chain for nickel, cobalt, and manganese in the case of China. These results, in addition to the LFP-Li and NMC-Li cases, are plotted in the leftmost set of five bars of Fig. 5, where bars represent the proportional case, error bars capture the range from the Optimistic Case to Pessimistic Case A, and the × represents the extreme Pessimistic Case B. In the proportional and minimum cases, the vulnerability index is highest for manganese [71% and 80%], while

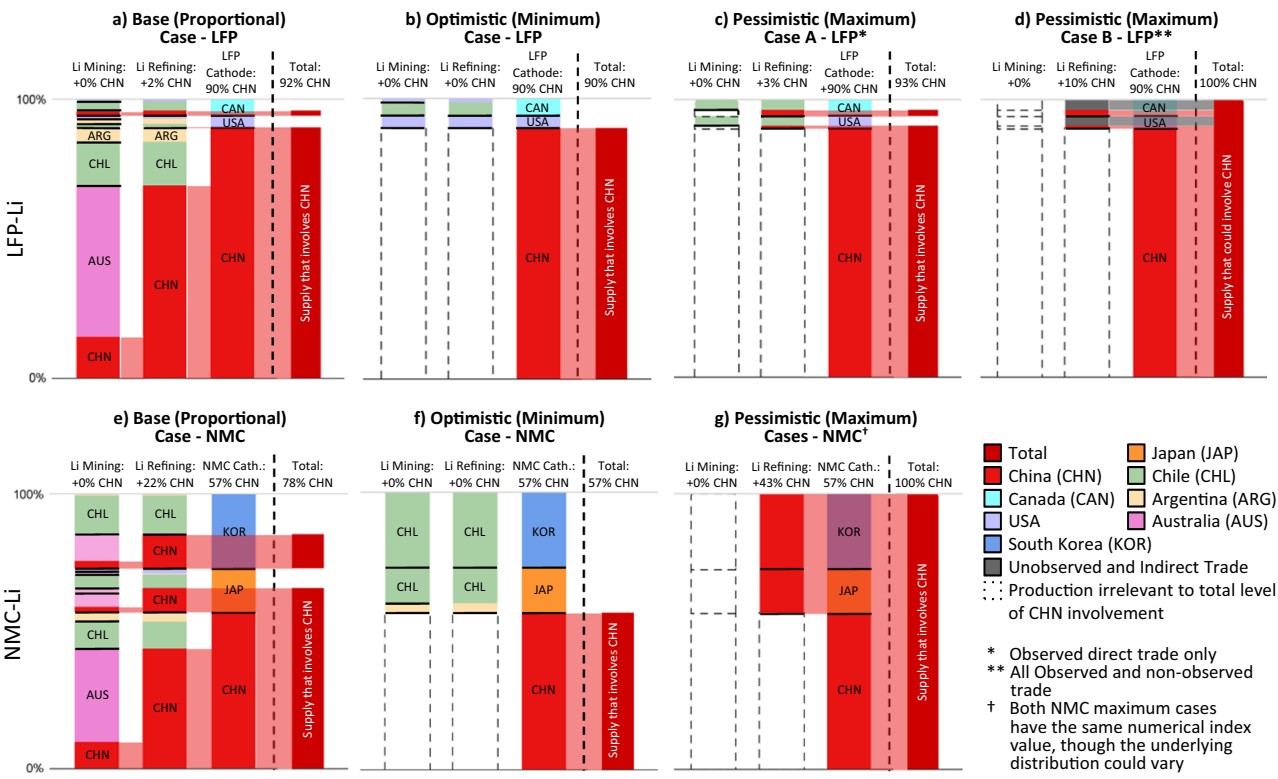

**Fig. 4 | Lithium vulnerability index for a supply chain disruption in China.** Visualization of a vulnerability index for global LFP (Lithium Iron Phosphate [a–d]) and NMC (Lithium Nickel Manganese Cobalt [e–g]) cathode supply, for a lithium supply chain disruption in China. Table 1 defines the four sensitivity cases. The level of overall vulnerability at the cathode manufacturing step, added at each upstream step, is noted with a plus (+) sign. Sums may not add due to rounding. The solid horizontal black lines are visual aids to indicate separation of countries; semi-transparent red bars represent vulnerability that is propagated downstream. The Li supply chains differ between LFP and NMC cathodes because of differences in the distribution of countries that produce each cathode and the supply chain paths of

each. Example solutions for the other NMC materials are in Supplementary Text S4-1. Indirect trade and unobserved trade do not count towards the measurement of vulnerability in the optimistic case, but are counted towards the pessimistic calculation in the case that includes uncertain data. For the minimum and maximum cases, one possible solution is presented, but other distributions with the same vulnerability index are possible. For all of these cases, we mask the distribution of production and trade upstream of China's supply at each supply chain step because it does not affect the vulnerability index and may be distributed arbitrarily in the network flow optimization results.

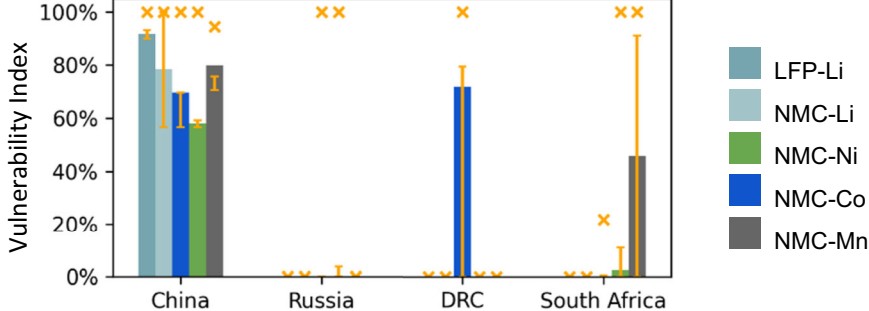

**Fig. 5 | Vulnerability index for chemistries LFP and NMC, materials Li, Ni, Co and Mn, and focal countries China, Russia, the Democratic Republic of Congo (DRC), and South Africa.** Bars show results using proportionality assumptions, error bars (in yellow) show range from optimistic (minimum) and pessimistic (maximum) based on uncertainty about intracountry flows, and the yellow × symbols show extreme upper bounds assuming all missing data originate from the

focal country. In the case of Russia, a substantial portion of the missing data for nickel and cobalt is known to originate in Russia[22,36,67]. We note that the vulnerability index in the proportional case for the NMC-Mn supply chain is greater than the vulnerability index in the pessimistic case that only includes visible trade (Pessimistic Case A). This is because a significant amount of refined manganese used in South Korea and Japan are from non-direct and non-observed trade (Fig. 3).

in the maximum cases the vulnerability index is highest for lithium [both 100%]. The numerical values presented in this figure are found in Table 2 (rounded to the nearest percent).

We aim to determine an overall vulnerability index for NMC as a material, but we suggest that any combination of the vulnerability indices presented here must be done carefully in order to avoid improper comparisons across minerals and chemistry choices. We assume a Leontief production function for multi-material systems like

NMC – namely that the mineral that has the lowest amount of supply (and thus highest vulnerability index) defines the overall vulnerability of the entire material system. We then compute an overall vulnerability index for NMC using the maximum vulnerability index across the four critical materials in each of the four cases presented.

Overall, then, LFP has a China vulnerability index of 92% [90% to 93%] due China's dominance in Li refining and LFP cathode production and NMC has a China vulnerability index of 80% [71% to 100%], due

**Table 2 | Summary of vulnerability index results for LFP and NMC battery chemistries for the case of China**

|  | Supply chain | Optimistic case (minimum) | Base case (proportional) | Pessimistic case A (known trade only) | Pessimistic case B (extreme bound) |
|---|---|---|---|---|---|
| **LFP** | Li | **90%** | **92%** | **93%** | **100%** |
|  | **Overall** | 90% | 92% | 93% | 100% |
| **NMC** | Li | 57% | 78% | **100%** | **100%** |
|  | Ni | 57% | 58% | 59% | **100%** |
|  | Co | 57% | 70% | 70% | **100%** |
|  | Mn | **71%** | **80%** | 76% | 94% |
|  | **Overall** | 71% | 80% | 100% | 100% |

Bold indicates that this material is the limiting factor for calculating vulnerability (i.e. the material with the highest vulnerability) for each chemistry. In Supplementary Text S1-2, we discuss phosphorus as a potential additional critical material for LFP.

most notably to China's dominance in NMC cathode production, Li refining, and Mn refining. In both cases a shift in cathode production away from China could reduce the vulnerability index to a limited degree, but a large reduction would require shifts in both cathode production and material refining. In particular, because NMC has four critical materials with high China vulnerability indices, shifts across the supply chains of all four materials would be needed to achieve large reductions in the vulnerability index. For both cases, the China vulnerability index is higher than what production concentration metrics alone would suggest. In Supplementary Text S1-2, we discuss how phosphorus might affect this comparison if it were to be considered a critical material.

Figure 5 also summarizes vulnerability indices for Russia, the Democratic Republic of the Congo (DRC), and South Africa. With proportionality assumptions, the vulnerability index is high for cobalt in the DRC and high for manganese in South Africa, but the bounding cases show substantial uncertainty – anywhere from 0% to 90% of the NMC supply chain is observed to be vulnerable to disruptions in Russia, the DRC and South Africa. The upper range can approach 100% given uncertainty of indirect trade and unobserved data: for example, the data suggest there is not much observed vulnerability to Russia, but if we were to assume that Russia supplies all of the unobserved nickel and cobalt supply in the battery materials supply chain, the vulnerability index could approach 100%. Of course, when the optimistic and maximally pessimistic vulnerability indices range from 0 to 100, this quantitative measure cannot serve as a precise measure of vulnerability. We suggest that such a calculation indicates that uncertainty is large for this supply chain, given available data, and thus if a policymaker is concerned about this combination of material and focal country, further investigation and detailed tracking of production and trade data would be warranted.

## Discussion

As vehicle electrification grows worldwide, a variety of decisions will impact firms' and countries' vulnerability to geopolitical and natural disruptions along the supply chain: not only where production happens and the number of firms and sites at each step of the supply chain, but also the chemistry of the battery itself. We mapped the current state of the four most critical battery materials, and then defined a disruption vulnerability index that quantifies, under uncertainty, the portion of LFP and NMC electric vehicle battery cathodes produced by or using materials from a given country. While the index captures disruption potential in current supply chains, not necessarily in future supply chains nor in concentrations of reserves that are not currently used for production, our method could be applied to any current multi-step, multi-material global supply chain. Our results present new

perspectives on uncertainty and geopolitical risk in the context of LFP, NMC, and their constituent critical minerals.

Our estimates indicate a high level of potential vulnerability when considering China's influence on the supply chain, with a vulnerability index of 92% [90% to 100%] and 80% [57% to 100%] for LFP and NMC, respectively. Though different countries and administrations may have varying thresholds of concern and risk tolerances, our methodology can help policymakers and researchers identify specific bottlenecks, key relationships, and potential levers for reduction of vulnerability or movement of production location in these interconnected supply chains, such as the key relationship between the Philippines and China regarding nickel or the dependence of South Korea and Japan on refined lithium from China. Using the diagrams, relationships, and analysis generated by this methodology, we hope to spur further research into country-specific vulnerabilities and the effects of specific disruptions on the electric vehicle battery material supply chain.

The uncertainty due to the observability of trade and missing trade data differs in important ways by chemistry, with Fig. 2, Fig. 3, and Fig. 5 shedding light on potential issues of data disclosure and supply chain transparency. The LFP-lithium supply chain has far less uncertainty in terms of unobservable or missing trade data than the supply chains for NMC: much of the production and trade data are observable for the entire lithium supply chain across material extraction, refining, and cathode production. In contrast, the non-lithium inputs into NMC have substantial uncertainty due to unobserved intranational flows, as well as both indirect and unobserved international trade. For example, China appears to source a substantial amount of refined cobalt from the DRC, but this amount of trade far exceeds the amount of refined cobalt production reported in the DRC. Other studies[22,36] indicate that unrefined cobalt undergoes initial processing within the DRC and further refining in China, as companies based in China own nearly all of the mines in the DRC[37]. Detailed investigation of our data indicates that nearly all of this traded refined cobalt is categorized as scrap, which also points to potential misclassification of materials as well as high uncertainty in the conversion factors used to calculate material in exports. This finding may help shed light on ongoing transparency and trade tracking issues in the cobalt supply chain, as few material flow analysis studies consider scrap trade in their assessments (see Supplementary Text S3-3). Furthermore, a substantial portion of battery-grade nickel (about 20% of worldwide supply) is known to be mined and refined in Russia (or neighboring Finland) and used in the European market[38,39], which is not reflected in the data either. These findings suggest that further emphasis should be placed on understanding how worldwide material production and trade is tracked, particularly around refined and scrap materials. Yet this knowledge of where the data is missing could be considered a further data point on the relative vulnerability when comparing between critical minerals - policymakers may want to choose to promote supply chains where the data is more well-understood and relative uncertainty is minimized.

These results may be unsurprising given the early and consistent support for electric vehicles and novel chemistries in China. Various studies in the literature[40–44] have identified a mix of national science and technology policy directives, government regulations, national and regional subsidies, market incentives and opportunities, and economies of scale (of both labor costs and manufacturing know-how) all contributing to China's currently dominating position in this industry. While lithium ion batteries were initially developed in the United States and commercialized in Japan, battery manufacturers largely started shifting towards China due to state subsidies and directives to invest in new energy technologies[42]. In 2000, the Chinese government initiated the "Going Out Strategy", encouraging companies to expand foreign direct investment for mineral resources in developing countries in Africa and Asia to support a burgeoning

domestic materials processing industry underlying the general manufacturing boom in the country, including battery materials like cobalt[44]. In the same year, the national government began massive investment in "New Energy Vehicles" along the entire technology pipeline of research to process commercialization as part of the 10th 5-year plan. Subsidies for electric vehicles started in 2009 that just ended in 2023, totaling in the hundreds of billions of Yuan (tens of billions of USD)[40,45]. This whole-supply chain approach provides a model for other countries to compare and contrast their domestic policy goals to if they are similarly concerned with building out their electric vehicle manufacturing and subsequent supply chains.

In the U.S. policy context, the Inflation Reduction Act[7] includes several provisions that encourage firms to change their location of production. A production tax credit equal to 10% of production costs incentivizes firms to domestically manufacture electro-active materials, such as the cathode or anode. In addition, EVs can qualify for two tax credits per vehicle, based on the geography of their supply chains: (1) those that have a minimum amount of critical minerals produced or processed domestically within free trade agreement countries and (2) a minimum amount of battery components manufactured in North America. Our results further suggest that these sourcing requirements may not avoid vulnerabilities due to specific geographies of concern along the entire supply chain (for both requirements). Furthermore, because of the compound nature of vulnerabilities across multiple supply chain stages, reduction of vulnerability at just one across these battery material supply chains is not sufficient.

A possible complementary policy action may be funding research, development, and demonstrations that aim to improve the performance of LFP batteries as well as the lithium extraction and refining processes, and thus reduce reliance on NMC and its compound material risks, as all four critical materials studied are required in order to produce NMC, creating multiple disruption paths. LFP also has an advantage given the relatively small overall physical quantity of materials that is present in the Lithium supply chain (comparing the scales in Figs. 2 and 3) and the fact that uncertainty appears to be the lowest in the Lithium supply chains, as described previously. Immense recoverable deposits of lithium are being rapidly discovered given the recent interest in the material[46–49]. In particular, the U.S. has substantial potential mining capacity (the Thacker Pass mine in Nevada is the third largest individual lithium resource in the world[50]), as well as existing capabilities in both mining and refining (Fig. 2). While choosing LFP may involve other technological and economic tradeoffs, a shift toward LFP may represent an opportunity to reduce disruption vulnerability, if lithium refining operations and LFP cathode production operations in particular are diversified away from the currently high concentration in China.

While the results presented have primarily focused on China as the single country with the largest influence on battery material supply chains and this vulnerability index, trade blocs or sets of politically aligned countries (e.g. a South American lithium production bloc) appear to also be able to have various levels of dominance on specific supply chains, as seen in Supplementary Text S4-2. The effect of other countries on multiple stages may not currently be nearly as pronounced, but the methodology generalizes our ability to understand such phenomena and the relative impact of any country or set of countries. Reframing vulnerability in a different context, this methodology could also allow policymakers to understand disruption vulnerability with respect to any set of countries and assess potential of alternative policy actions to affect overall supply chain vulnerability. Our results thus suggest that because individual countries and trade blocs can dominate multiple stages of the supply chain, reducing battery supply chain risks (or strengthening networks of trade between friendly nations) requires understanding and addressing not just immediate but secondary and tertiary vulnerabilities.

## Methods

A full description of the methodologies and data used to develop this analysis are described in Supplementary Text S3. We simplify a fully circular supply chain model in our model, which is presented in Supplementary Text S1-1.

We first build on the methodologies of the Material Flow Analysis literature[16,21–25,36,51–56] to map and characterize the production and flow of materials in battery supply chains in a "data aggregation" step. To do so, we begin with the amount of material known to be produced in each of the mining, refining, and cathode production steps. Each country can produce, import, and export multiple materials at each step of the supply chain, so we convert production and trade statistics into quantities of contained materials. We choose to use reported import data quantities as it tends to be more complete and accurate in comparison to reported export data[22,23,57,58]. For example, we convert trade quantities in nickel ores and concentrates, nickel mattes, nickel sulfates, nickel waste and scrap, etc. into units of contained nickel, as prior literature[27,56,59–62] indicates that some or all of these trade flows may contain nickel material that ultimately is used in battery manufacturing. See Supplementary Text S3 for further discussion on the chosen data sources and trade codes, and corresponding literature. We then balance the amount of each contained material between the mining step and refining step, and between the refining step and the cathode manufacturing step, incorporating trade at each stage.

We then use these flows to calculate the total vulnerability index for cathode material supply for a focal country or set of countries. We estimate our base case assuming the distribution of inputs for each country at each stage flows proportionally to the outputs, in line with ideas of input-output models[63] or the supply chain disruption propagation literature[29]. To bound the vulnerability index, given uncertainty about intracountry flows, we model the supply chain as a network flow problem and optimize the uncertain intracountry flows to minimize or maximize the portion of cathode production using materials that pass through the focal country[64]. To map our network structure to the network flow problem, connections between nodes are assigned capacity values equal to flows calculated through our data aggregation step, and all mining and mineral extraction is connected to a supersource node. This supersource node is also connected (with infinite capacity) to each country of focus' node(s) along the steps of the supply chain, so that the maximal amount of flow from this supersource to the terminal node can be measured. In the maximal case with uncertain data, the nodes representing uncertain sources of materials are also connected to the supersource. In the minimal case, all connections involving any countries of focus are disconnected from the network, and the maximal flow is calculated. The difference between this quantity and the amount of material demanded is the minimum vulnerability index for the countries of focus. A detailed formulation is provided in Supplementary Text S3-1.

We note that this methodology is generally flexible and can be augmented with further data as well as applied to any multi-step, multi-material global supply chain given sufficient production and trade data. For example, this study only includes materials scrap trade codes as some measure of circularity. While current trade of end-of-life batteries and battery material supplies is minimal, if trade codes are established for such materials, or trade of such materials could be sufficiently estimated from future trade codes, this methodology could easily incorporate that additional data stream. Furthermore, we believe this methodology would be readily applicable in other supply chains, such as battery chemistries that are not yet on the market, rare earth minerals for motors, turbines, electrolyzers, solar panels, and the like, given sufficient production and trade data.

### Reporting summary

Further information on research design is available in the Nature Portfolio Reporting Summary linked to this article.

## Data availability

Source data are provided with this paper; we only use publicly available production and trade data for this analysis, from the U.S. Geological Service's Mineral Commodity Survey[17], Sun et al. 2021[59], International Fertilizer Association[65], and IntraCen's TradeMap[57], which is described in detail in Supplementary Text S3-2. Data Sources. The source and processed data generated in this study have been deposited in a public GitHub repository with no accession code (https://doi.org/10.5281/zenodo.10607313)[66].

## Code availability

The code that is used for this analysis is freely available in the Github repository linked to this paper (https://doi.org/10.5281/zenodo.10607313)[66].

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

## Acknowledgements

This research was supported by the 2021 Carnegie Mellon University College of Engineering Moonshot Award for "Engineering Competitiveness: Critical Technologies, Supply Chains, and Infrastructure," the National Science Foundation Award # 2241237, and the National Science Foundation Graduate Research Fellowship under Grant # DGE2140739. We would like to acknowledge Paulina Jaramillo, Elsa Olivetti, Katie Whitefoot, and Jared Cohon for feedback that helped us improve initial drafts of this article, the Fuchs, Karplus (LEO), and Michalek (VEG) research groups for their feedback throughout the development of this work, and Ryan Liu for his discussions about maximum flow algorithms. We acknowledge the use of ChatGPT as a support tool to generate code snippets and help debug code issues.

## Author contributions

CRediT authorship contributions: Anthony Cheng: Conceptualization, Methodology, Software, Formal analysis, Writing - original draft, Writing - review & editing, Visualization. Erica Fuchs - Conceptualization, Methodology, Writing - review & editing, Visualization, Supervision, Funding acquisition. Valerie Karplus - Conceptualization, Methodology, Writing - review & editing, Visualization. Jeremy Michalek - Conceptualization, Methodology, Writing - review & editing, Supervision, Visualization.

## Competing interests

The authors declare no competing interests.
