## [Peer Review File · Nature Communications]

Electric vehicle battery chemistry affects supply chain disruption vulnerabilitiesReviewers' Comments:

Reviewer #1:

Remarks to the Author:

This is a well-structured paper that develops an index for measuring the supply chain vulnerability for the two dominant Li-ion battery chemistries (NMC and LFP) associated with electric vehicles. It offers a novel method for calculating this index and provides results for supply chain vulnerabilities associated with China's aggregate role across upstream, midstream and downstream stages of the supply chain. The main innovation lies in combining multiple stages of the supply chain (which shows in the case of China that vulnerabilities are higher than when calculated for only individual stages) with a granular approach to building the model that tracks the geography of the supply chain back from cathode material (and adjusts for uncertainties in trade data) rather than using national production figures alone.

The paper is logical and clearly presented for the most part. Claims are sufficiently supported with evidence and supported by accompanying materials including effective graphics. There is scope (as outlined below) for further interpretation and comment on the meaning and significance of the findings. There are no substantial flaws as far as I can see but there are some points for clarification: some are relatively minor while others involve further reflection on the relationship between the general methodology of a disruption vulnerability index described in the paper and the specific case (EV battery minerals, with China as the focal country) chosen to illustrate it.

Overall, the paper is likely to generate discussion around the range of methodologies for assessing supply chain risk. The literature review is good on this point and the effort to map the paper against '6 primary qualities' helps clarify the distinctiveness. The difference between the authors' index and criticality assessments is important and more could be made of this in the main texts. While this index and criticality studies both adopt a 'focal country' for their frame of reference, criticality assessments adopt the geography of the importing country (and assess import dependence and the significance of the mineral to the national economy) while the authors calculate their index around the geography of a battery mineral supplier (assessing the contribution of materials extracted, refined and manufactured in that country to the overall supply chain). In short, this difference of 'focal country' means they adopt quite different perspectives on the problem of vulnerability to supply disruption.

The uncertainties around trade data are described as falling into three categories. How big are these uncertainties overall? These uncertainties seem important – particularly the unobserved flows – as they go to the heart of the paper's effort to trace the supply chain (rather than use only national aggregate figures of production/capacity). Perhaps this is mentioned somewhere, but I was unable to see some assessment of how large a contribution these estimated figures make to the total: the method for dealing with them is described, but it would be useful to know how much of the aggregate trade flows are attributable to these 'uncertain' data and the proportionality assumptions that were applied – 50%, 10%, 1%? On page 9, line 221, in relation to China these uncertainties are described as 'significant'.

The findings described are largely technical, in the sense they are about the inputs to and outputs from the model, and there is relatively little reflection on the historic or contemporary international political economy of battery production. In this context, the primary finding – that LFP has higher supplier chain vulnerability than NMC – is not that surprising, given that LFP chemistry until relatively recently been largely a preserve of the Chinese EV market (it has shifted quite rapidly in the last couple of years). Given that the paper sets itself up as being about geopolitical economy of EV battery minerals, there is scope for a bit more interpretative reflection of what lies behind the findings (e.g., 30 years or so of neoliberal trade policy, which saw production of batteries and battery minerals 'outsourced' to China, combined with – in the last 20 years – a strategic move by China to dominate the midstream stages in this supply chain). More generally, what should we make of the fact that LFP has an index of 92% and NMC has one of 80%? What is the meaningful implication of this difference?

At a basic level, both read as 'high' – is there an imagined threshold for concern here that readers should be aware of? Given the substantial time and resources likely allocated to the calculative modelling in the paper, it seems non-trivial to ask whether the outcome – i.e., knowing that NMC is 80% vs. LFP at 92% - is meaningful. I don't say this to be dismissive: it is about knowing what kind of weight the authors think should be placed on this finding, and to what sort of actions it should give rise.

The contribution of second-life materials to the supply chain, from the recovery and recycling of end-of-life EV batteries, does not seem to be a feature of the analysis. The volumes available have, up until this point, been largely insignificant (other than for cascading uses – e.g., in static energy storage). However, there is increasing investment in materials recovery – and initiatives like the Global Battery Passport and circular economy in the EU – which have the potential to alter the geography of battery mineral availability: indeed, they are designed strategically to do precisely this. In this sense, the linear 'supply chain' approach at the heart of the model seems rather inflexible: to what extent can it accommodate (and distinguish) circular economy models and the growing availability of 'end-of-life' lithium supplies?

Figures 2 and 3 are useful and sufficiently clear. Why did you choose (p.16, line 381) to "convert observed trade quantities in nickel ores and concentrates, nickel mattes, nickel sulfates, etc. into units of contained nickel" rather than focusing on nickel sulphate alone, given it is nickel sulphate that enters the battery supply chain? Figure 4 is complex but sufficiently clear.

There is some tension in the paper between proposing a general index approach, that might be applied to any one of a number of countries or metals, and the specific case of China and EV battery minerals that is used to illustrate the case. China occupies a unique position with regard to its dominance in the midstream and downstream stages of the lithium-ion battery production chain. The 'multiple stages' argument in the paper, then, works for China (e.g., p.12: 'the vulnerability index is higher than what production concentration metrics alone would suggest') but looks much more limited for other countries. The authors' conclusion on p.8 in relation to cobalt, nickel and manganese seems to recognise this, as they note that "even though large producers such as Indonesia, Australia, and Gabon exist, the amount of influence they have on battery material supply chains may be limited." On the other hand, it is not clear why the methodology proposed here should be restricted to supply chain vulnerabilities in relation to battery minerals: what, for example, about a wider suite of e-tech minerals, including rare earths and platinum group metals, for example – could not this methodology be also applicable to them? In other words, the particularity of the case (EV/China) and the generality/general applicability of the case merit some further consideration.

Reviewer #2:

Remarks to the Author:

The authors present an analysis on material flows by country for li-ion EV battery cathodes (LFP, NMC) to develop vulnerability index ratings for cathode critical materials. The work is timely, integrates multiple approaches for assessing vulnerabilities, and generates new analysis around sources of uncertainty. My comments are primarily around communication and visualizations.

* Appendix A1: The figure is helpful, but a table format is confusing... maybe consider visualizing this as a process diagram that shows the different elements of the supply chain, and what's included in the these different levels of control. A general process diagram of the supply chain would also be helpful in providing context for the reader, which could go in this same Appendix.

* Line 39: I think it's important to mention the more local economic motivation behind these laws/regulations beyond supply chain resilience - the drive to realize the more local social and economic benefits around domestic production in place of globalization.

* Fig. 1: Great figure!

- * 137: A clear statement on what this study does to improve this detailed body of literature would be helpful. How does this study synthesize these different approaches and then improve on each area individually, or is the approach integration a novel contribution in combination with specific improvements?
- * Fig 2: Can you include the scale in the legend? This comment goes for the following Sankeys as well. For example see Heller et al. 2020 for a US Sankey on plastics with a scale legend.
- * Fig 3: Recommend moving the "b)Nickel" label to just below the dashed line, and including a dashed line to separate b) from c)
- * A lot of the results is concentrated in the MFA - which it sounded like was not the novel piece of the analysis from the Lit Review description.
- * 202: missing a colon in #2 after "data"
- * Fig 4: It looks like Fig. 4 is like a condensed version of a handful of Sankey diagrams, which given the space constraints for the journal makes sense. It might be helpful to provide a full Sankey of one of these sensitivity scenarios in the Appendix to show some of It would be helpful to provide some clear context and takeaways in the discussion prior and after the figure to help the reader with the large amount of information condensed in this figure. For example, adding another sentence in the paragraph in 230 around the key takeaway, namely that if there is disruption in China, 90% of the LFP global supply chain is vulnerable to disruption (i.e., the vulnerability index result).
- * Fig. 4: Why is the mining and refining of NMC Li so different than LFP? Is it just because China sources more in-country lithium for NMC?
- * Fig. 4: figure f) has "LFP Cathode" in the x-axis label
- * 364: Could you add any points on cathode recycling and recovery? Are there alternative cathode materials that are use significantly less critical materials?
- * Discussion: How do you start to dive deeper into these numbers? For example, it's clear these supply chains are very concentrated in China, and because of that distribution they are more vulnerable. Moving forward, is it possible to make this type of analysis more robust by adding how vulnerable the country is? Is it mores than another country? What are the drivers (climate change, infrastructure, resources, political unrest)?

Reviewer #3:

Remarks to the Author:

Review of: Electric vehicle battery chemistry affects supply chain disruption vulnerabilities

Overall assessment:

This article provides an in depth study of the vulnerability of disruption in the EV battery industry due to disruption in the supply in the material systems. To this end, the authors define a disruption vulnerability index which is assessed for each material at every stage of the material manufacturing processes and use a network flow optimization model to account for the uncertainties in the data.

While this article provides some quantification to the risk of disruption of some battery materials, the ranges provided are often so large that the conclusions become insignificant. In several instances, it is mentioned that the range is 0% to 100% (II. 249-250, II. 294-296, II. 326-328), which does not generate any valuable information since risk is always within this range by definition. The wide ranges preented can be expected, as it is well known that trade data are not fully harmonized and it can become difficult to assess how much a material is contained in each trade code. A particular concern for this reviewer are assumptions underlying the trade codes used to identify the battery chemistry of the batteries being traded, which is a central issue to this paper: Trade codes are not disaggregated enough to differentiate the battery chemistry of EVs, they are merely reported based on their drive train. It is therefore unclear how the authors managed to differentiate an LFP EV from a NCM EV.

Another major concern I have is the active decision to not account for phosphorus (P) in the vulnerability assessment of batteries. P is a key material for LFP batteries and is presented as a better

alternative than Ni and particularly Co, since it is not mined in the DRC. While this is true, P is considered a critical material not only as phosphorus itself, but also as elemental phosphorus (P₄) used in LFP batteries, since their supply chains are extremely concentrated and the resource may compete with the fertilizer industry used for food production (see also <https://www.nature.com/articles/s43246-022-00236-4>). Ignoring the risks linked to P in LFP batteries is neglecting a massive part of the potential risks in the production of LFP cathode materials and the conclusions that can be drawn are therefore incomplete. It is therefore the opinion of this reviewer that the comparison presented in this paper is only partially completed and the results presented can therefore be misleading.

For the major reasons outlined above, I would warmly recommend to the authors to include P and P₄ in their assessment, and to be more transparent about the trade codes used and their limitations in their assessment. Until this is the case, I cannot recommend that this article be published in this prestigious journal.

Specific comments

1. Structure: I find it rather confusing that you have split the introduction into background and literature review. It makes it hard to follow and you find information addressing the same topic in different places. I would encourage you to merge these sections into one logically-constructed introduction.
2. Ll. 95-98: References?
3. Ll. 106-107: What do you mean with replicate the methods? This sentence is unclear to me.
4. Ll. 147-149: Why did you only consider the LIB industry? The materials are used in various applications and so disruptions can be caused as much by LIB demand as by the other user industries. Also, very importantly here, which trade codes did you use? As far as I am aware, no separate trade codes exist for battery chemistries. Did you use import or export data?
5. Ll. 151-153: What about stocks? The concentration process of lithium brines takes sometimes years and so the mining is not equal to the available Li in a given year. Also, Co is notoriously known for its opaque supply chains and black market. How can this be mass balanced if the data are not comprehensive?
6. Section "Measuring a Vulnerability Index": Import and export data often disagree with each other. How was this issue treated? How did you deal with double-counting?
7. Ll. 214-216: This is a big assumption as some countries are known to be better reporters than others. How can this be justified?
8. Figure 4: Do the proportions of risk assessment presented here make sense? It seems like you just added the vulnerability index at each stage for the total risk. Shouldn't it be a weighted average?
9. Ll. 248: Which data did you use for this? Sources?
10. Figure 5: It seems that the aggregation of vulnerability indices can exceed 100%. Why and how can this be the case?
11. Ll. 312-319: This is only repeating the results. Consider either putting it in context or removing.
12. Ll. 320-334: This entire paragraph is very descriptive but it is difficult to extract the relevance and insight that your points here generate. Consider making the text more targeted towards the relevance rather than the crude facts.
13. Ll. 340-343: Doesn't this indicate that the trade data on cobalt is too uncertain to be useful?
14. Ll. 349-363: This is common knowledge and general statements that don't follow from your results. How does your study build on this knowledge base?
15. Ll. 364-367: I don't understand this recommendation, wouldn't this just give China even more control? Neglecting P completely skews the understanding of LFP disruption risks as well. I don't think this statement is substantiated unless a thorough analysis of P is also conducted.

Response to Reviewer Comments

for Nature Communications

“Electric vehicle battery chemistry affects supply chain disruption vulnerabilities”

by Anthony Cheng, Erica Fuchs, Valerie Karplus and Jeremy Michalek

We thank the three reviewers for their valuable comments, which have helped us improve the manuscript. Below we respond point-by-point to each reviewer’s comments in blue text, indicating where we have made changes to the revision to address each comment.

Reviewer #1

This is a well-structured paper that develops an index for measuring the supply chain vulnerability for the two dominant Li-ion battery chemistries (NMC and LFP) associated with electric vehicles. It offers a novel method for calculating this index and provides results for supply chain vulnerabilities associated with China’s aggregate role across upstream, midstream and downstream stages of the supply chain. The main innovation lies in combining multiple stages of the supply chain (which shows in the case of China that vulnerabilities are higher than when calculated for only individual stages) with a granular approach to building the model that tracks the geography of the supply chain back from cathode material (and adjusts for uncertainties in trade data) rather than using national production figures alone.

The paper is logical and clearly presented for the most part. Claims are sufficiently supported with evidence and supported by accompanying materials including effective graphics. There is scope (as outlined below) for further interpretation and comment on the meaning and significance of the findings. There are no substantial flaws as far as I can see but there are some points for clarification: some are relatively minor while others involve further reflection on the relationship between the general methodology of a disruption vulnerability index described in the paper and the specific case (EV battery minerals, with China as the focal country) chosen to illustrate it.

Thank you for your thorough assessment and helpful recommendations.

Overall, the paper is likely to generate discussion around the range of methodologies for assessing supply chain risk. The literature review is good on this point and the effort to map the paper against ‘6 primary qualities’ helps clarify the distinctiveness.

1. The difference between the authors’ index and criticality assessments is important and more could be made of this in the main texts. While this index and criticality studies both adopt a ‘focal country’ for their frame of reference, criticality assessments adopt the geography of the importing country (and assess import dependence and the significance of the mineral to the national economy) while the authors calculate their index around the geography of a battery mineral supplier (assessing the contribution of materials extracted, refined and manufactured in that country to the overall supply chain). In short,

this difference of 'focal country' means they adopt quite different perspectives on the problem of vulnerability to supply disruption.

We appreciate the reviewer's observation that most studies consider imports and demand as the primary viewpoint for criticality assessments, while our study includes the geography of supply. We've added language to clarify that both import/demand and export/supply perspectives are important in this analysis, on p. 5 as follows:

"We suggest there is an important gap in the existing literature given a lack of analysis of the vulnerabilities in the global flow of multiple battery materials between countries as sources of both supply and demand across stages of the supply chain, particularly in terms of physical quantities of materials. While the criticality literature typically takes the perspective of an importing focal country, our study instead assesses global supply chain dependence on exporting focal countries. Moreover, existing studies do not explore multiple sources of uncertainty. While we do not attempt to determine potential causes of risks and vulnerabilities, nor assess the probability of such risks or vulnerabilities occurring, we aim to provide a quantitative metric to help assess vulnerabilities and evaluate potential actions to reduce vulnerabilities. In addition to specific insights about the supply chains for electric vehicle batteries, including the explicit inclusion of the cathode material manufacturing step, we provide a novel methodology incorporating material flow analysis methods with optimization methods to account for the impact of uncertainty on measurements of vulnerability."

The uncertainties around trade data are described as falling into three categories. How big are these uncertainties overall? These uncertainties seem important – particularly the unobserved flows - as they go to the heart of the paper's effort to trace the supply chain (rather than use only national aggregate figures of production/capacity).

2. Perhaps this is mentioned somewhere, but I was unable to see some assessment of how large a contribution these estimated figures make to the total: the method for dealing with them is described, but it would be useful to know how much of the aggregate trade flows are attributable to these 'uncertain' data and the proportionality assumptions that were applied – 50%, 10%, 1%? On page 9, line 221, in relation to China these uncertainties are described as 'significant'.

In the revision we clarify what is known about the contribution of each source of uncertainty. We cannot break down the contribution of the sources of uncertainty linearly because they interact in nonlinear ways, so instead we discuss their levels of contribution. The revised text, on p. 9 and p. 10, reads as follows:

"The most simple method to measure a vulnerability index, in line with ideas from the supply disruption propagation and input-output literature, is to calculate a

point estimate “proportional” case. To do this, we only take into account the proportional share of imports and domestic production for determining the relationship of each country with the upstream step. Results for the proportional LFP and NMC analyses are summarized in Fig. 4(a,e). For LFP, 90% of cathode production takes place in China, and the US and Canada make up the remaining 10%, suggesting a vulnerability index of at least 90%.”

“Using only the trade data that is present and accounting for the first type of uncertainty (unobserved intranational flows), we estimate minimum and maximum bounds, as for example we do not necessarily know where the observed production of LFP in China sources lithium from. This corresponds to an optimistic case and pessimistic case A. With our base case proportional flow estimates [optimistic estimate to pessimistic case A estimate], we estimate that 17% [0% to 33%] of refined Li imports to the US and Canada are refined in China, increasing the vulnerability index by +2 [+0 to +3] percentage points to a total of 92% [90% to 93%] of cathode material production involving China, depending on which US and Canada imports are used for LFP cathode production. However, it is important to recognize that this scenario, while pessimistic, is not a guaranteed upper bound, as uncertainty may exist due to non-observed and indirect trade that could be attributed to production in China (uncertainties 2 and 3). As a result we include a second, more extreme upper bound (Pessimistic Case B) by simultaneously assuming that all indirect trade and missing materials originate from China, resulting in the vulnerability index for LFP increasing by +7 percentage points when comparing the two pessimistic cases to 100%, as seen in Fig. 4(d). These results suggest that LFP battery material supply chains are highly vulnerable to a disruption in China, as even in the most optimistic case the vulnerability index is over 90%. While not noted on the figure, further analysis indicates that even if all cathode production were moved out of China, an estimated 71% of LFP cathode material would use Li inputs produced or traded from China in the proportional case. ”

3. The findings described are largely technical, in the sense they are about the inputs to and outputs from the model, and there is relatively little reflection on the historic or contemporary international political economy of battery production. In this context, the primary finding - that LFP has higher supplier chain vulnerability than NMC – is not that surprising, given that LFP chemistry until relatively recently been largely a preserve of the Chinese EV market (it has shifted quite rapidly in the last couple of years). Given that the paper sets itself up as being about geopolitical economy of EV battery minerals, there is scope for a bit more interpretative reflection of what lies behind the findings (e.g., 30 years or so of neoliberal trade policy, which saw production of batteries and battery minerals ‘outsourced’ to China, combined with - in the last 20 years - a strategic move by China to dominate the midstream stages in this supply chain). More generally, what should we make of the fact that LFP has an index of 92% and NMC has one of 80%? What is the meaningful implication of this difference? At a basic level, both

read as ‘high’ – is there an imagined threshold for concern here that readers should be aware of? Given the substantial time and resources likely allocated to the calculative modelling in the paper, it seems non-trivial to ask whether the outcome – i.e., knowing that NMC is 80% vs. LFP at 92% - is meaningful. I don’t say this to be dismissive: it is about knowing what kind of weight the authors think should be placed on this finding, and to what sort of actions it should give rise.

We fully agree that there is greater latitude to reflect on the historical backdrop and use this to contextualize our findings and examine their implications. On the first issue, involving history and trade policy, in the revision we cite 6 papers that summarize this history with the following new text, copied below, provided starting on p. 15:

“These results may be unsurprising given the early and consistent support for electric vehicles and novel chemistries in China. Various studies in literature^{40–44} have identified a mix of national science and technology policy directives, government regulations, national and regional subsidies, market incentives and opportunities, and economies of scale (of both labor costs and manufacturing know-how) all contributing to China’s currently dominating position in this industry. While lithium ion batteries were initially developed in the United States and commercialized in Japan,⁴² battery manufacturers largely started shifting towards China due to state subsidies and directives to invest in new energy technologies. In 2000, the Chinese government initiated the “Going Out Strategy”, encouraging companies to expand foreign direct investment for mineral resources in developing countries in Africa and Asia to support a burgeoning domestic materials processing industry underlying the general manufacturing boom in the country, including battery materials like cobalt.⁴⁴ In the same year, the national government began massive investment in “New Energy Vehicles” along the entire technology pipeline of research to process commercialization as part of the 10th 5-year plan. Subsidies for electric vehicles started in 2009 that just ended in 2023, totaling in the hundreds of billions of Yuan (tens of billions of USD).^{40,45} This whole-supply chain approach provides a model for other countries to compare and contrast their domestic policy goals to if they are similarly concerned with building out their electric vehicle manufacturing and subsequent supply chains.”

On the second issue, involving significance and implications of the findings, we’ve added additional language in the Conclusions and Policy Implications section, on p. 14, to address our views on this, copied below for convenience. Specifically, we view it as not our role as scholars to establish thresholds, since doing so involves risk assessments and risk preferences that may vary across countries and administrations. Rather, we see our index as helping to provide quantitative information to decision-makers about the structure and degree of vulnerability, including an assessment of what kinds of actions could reduce the vulnerability index and what kinds would not. For example, with our index one can see that shifting cathode production to other countries without shifting

material refining operations may have limited overall effects of protecting the supply chain overall from a potential disruption in China.

“Our estimates indicate a high level of potential vulnerability when considering China’s influence on the supply chain, with a vulnerability index of 92% [90% to 100%] and 80% [57% to 100%] for LFP and NMC, respectively. Though different countries and administrations may have varying thresholds of concern and risk tolerances, our methodology can help policymakers and researchers identify specific bottlenecks, key relationships, and potential levers for reduction of vulnerability or movement of production location in these interconnected supply chains, such as the key relationship between the Philippines and China regarding nickel or the dependence of South Korea and Japan on refined lithium from China. Using the diagrams, relationships, and analysis generated by this methodology, we hope to spur further research into country-specific vulnerabilities and the effects of specific disruptions on the electric vehicle battery material supply chain.”

4. The contribution of second-life materials to the supply chain, from the recovery and recycling of end-of-life EV batteries, does not seem to be a feature of the analysis. The volumes available have, up until this point, been largely insignificant (other than for cascading uses – e.g., in static energy storage). However, there is increasing investment in materials recovery – and initiatives like the Global Battery Passport and circular economy in the EU – which have the potential to alter the geography of battery mineral availability: indeed, they are designed strategically to do precisely this. In this sense, the linear ‘supply chain’ approach at the heart of the model seems rather inflexible: to what extent can it accommodate (and distinguish) circular economy models and the growing availability of ‘end-of-life’ lithium supplies?

We agree that a linear supply chain modeling approach is inflexible and would not be able to account for things like increases in recycling or scrappage. However, our approach is different and more flexible. We have included additional language on p. 6 of the Results section, as well as the beginning and end of the methodology section, on p. 18 and p. 19, to specify that scrap materials were included in the original analysis and how further analyses could extend upon the work presented in this paper given future data streams:

Results: “We describe the global supply chains for lithium in Fig. 2 and for cobalt, nickel and manganese in Fig. 3., considering the known demand for various lithium ion battery cathode materials. From left to right, each figure shows the portion of global material supply in each country for raw material mining and extraction, raw material trade, refining and processing, refined material trade, and cathode production, which includes trade of scrap or recovered materials that may be involved in the battery material supply chain. ... Details about the methodology and data underlying in Fig. 2 and Fig. 3 are found in the

Methodology section and Appendix C, with the simplified version of the supply chain modeled in this paper presented in Appendix A1.”

Methodology: “A full description of the methodologies and data used to develop this analysis are described in the Appendix C, with the simplified version of the supply chain modeled in this paper presented in Appendix A1.”

“We note that this methodology is generally flexible and can be augmented with further data as well as applied to any multi-step, multi-material global supply chain given sufficient production and trade data. For example, this study only includes materials scrap trade codes as some measure of circularity. While current trade of end-of-life batteries and battery material supplies is minimal, if trade codes are established for such materials, or trade of such materials could be sufficiently estimated from future trade codes, this methodology could easily incorporate that additional data stream. Furthermore, we believe this methodology would be readily applicable in other supply chains, such as battery chemistries that are not yet on the market, rare earth minerals for motors, turbines, electrolyzers, solar panels, and the like, given sufficient production and trade data.”

5. Figures 2 and 3 are useful and sufficiently clear. Why did you choose (p.16, line 381) to “convert observed trade quantities in nickel ores and concentrates, nickel mattes, nickel sulfates, etc. into units of contained nickel” rather than focusing on nickel sulphate alone, given it is nickel sulphate that enters the battery supply chain? Figure 4 is complex but sufficiently clear.

We appreciate the positive comments on the clarity and usefulness of the Figures. Additional language has been added after the quoted line, on p. 17, to clarify this statement on nickel specifically (below) as well as in Appendix C3.

“For example, we convert observed trade quantities in nickel ores and concentrates, nickel mattes, nickel sulfates, nickel waste and scrap, etc. into units of contained nickel, as prior literature^{18,28,56,59–61} indicates that some or all of these trade flows may contain nickel material that ultimately is used in battery manufacturing.”

6. There is some tension in the paper between proposing a general index approach, that might be applied to any one of a number of countries or metals, and the specific case of China and EV battery minerals that is used to illustrate the case. China occupies a unique position with regard to its dominance in the midstream and downstream stages of the lithium-ion battery production chain.
The ‘multiple stages’ argument in the paper, then, works for China (e.g., p.12: ‘the vulnerability index is higher than what production concentration metrics alone would suggest’) but looks much more limited for other countries. The authors’ conclusion on p.8

in relation to cobalt, nickel and manganese seems to recognise this, as they note that “even though large producers such as Indonesia, Australia, and Gabon exist, the amount of influence they have on battery material supply chains may be limited.”

We appreciate the point that the results primarily discuss the impact of China, and China seems to be, presently, the only country that may directly influence multiple stages of the supply chain. We’ve added additional language, starting on p. 16, to highlight other scenarios presented in Appendix D. Figure D2-2 that involve other trade blocs or geopolitically-aligned countries.

“While the results presented have primarily focused on China as the single country with the largest influence on battery material supply chains and this vulnerability index, trade blocs or sets of politically aligned countries (e.g. a South American lithium production bloc) appear to also be able to have various levels of dominance on specific supply chains, as seen in Appendix D2. The effect of other countries on multiple stages may not currently be nearly as pronounced, but the methodology generalizes our ability to understand such phenomena and the relative impact of any country or set of countries. Reframing vulnerability in a different context, this methodology could also allow policymakers to understand disruption vulnerability with respect to any set of countries and assess potential of alternative policy actions to affect overall supply chain vulnerability. Our results thus suggest that because individual countries and trade blocs can dominate multiple stages of the supply chain, reducing battery supply chain risks (or strengthening networks of trade between friendly nations) requires understanding and addressing not just immediate but secondary and tertiary vulnerabilities.”

7. On the other hand, it is not clear why the methodology proposed here should be restricted to supply chain vulnerabilities in relation to battery minerals: what, for example, about a wider suite of e-tech minerals, including rare earths and platinum group metals, for example – could not this methodology be also applicable to them? In other words, the particularity of the case (EV/China) and the generality/general applicability of the case merit some further consideration.

We appreciate the consideration that this methodology can be applied in a more general sense beyond the scope of electric vehicles; additional language to that effect has been added to the Conclusions and Policy Implications and Methodology sections, noting that this was combined with the response to point #4.

Reviewer #2

The authors present an analysis on material flows by country for li-ion EV battery cathodes (LFP, NMC) to develop vulnerability index ratings for cathode critical materials. The work is timely, integrates multiple approaches for assessing vulnerabilities, and generates new analysis around sources of uncertainty. My comments are primarily around communication and visualizations.

1. Appendix A1: The figure is helpful, but a table format is confusing... maybe consider visualizing this as a process diagram that shows the different elements of the supply chain, and what's included in these different levels of control. A general process diagram of the supply chain would also be helpful in providing context for the reader, which could go in this same Appendix.

We appreciate the reviewer suggestion that a table may be confusing for representation of the supply chain. In the revision we have moved what was previously Appendix D's supply chain diagram into Appendix A and incorporated the supply chain control steps into the new Figure A1-2. A reworked version of Table A1-1 is included as well, with additional text included to clarify the general process diagram of the supply chain. These changes are copied below for convenience.

Fig. A1-2: The simplified supply chain flows for battery materials considered in this study (in solid or dashed black lines), with manufacturer firm boundaries as described in Table A1-1. Note the steps in light gray are not modeled in this analysis.

Table A1-1. EV manufacturer decision making in their supply chain. Arrows indicate general movement of firms towards increasing direct control over their supply chains. Adapted and augmented from^{66,67} and company press releases

Supply Chain Control	Examples
Low Battery control: Cell, module, and pack outsourced to 3rd party supplier, usually a battery manufacturer (e.g. CATL)	Many EV startups: NIO, Lucid, Fisker, etc.
Moderate Battery control: Cell production outsourced. In-house module and pack design and manufacturing	BMW, Renault, Daimler, VW↓
High Battery control: Cell production through joint ventures/ partnerships, and/or in-house module and pack design and manufacturing	Nissan, Mitsubishi, PSA (Stellantis), Toyota, Geely / Volvo, GM↓,
Total Battery control: Firm manufacturing of cell, module, and pack design	BYD↓, Tesla↓, Ford↓
Battery control and high supply chain integration: Firm raw materials processing	
Battery control and complete integration: Integrated mining and raw materials extraction	

By definition, all companies producing their own battery cells, either in-house or through joint ventures or partnerships, are actively deciding the battery chemistries they have in their vehicle batteries, and thus where they source their battery cathode material from and the risks they might face due to global trade. Additionally, some companies (e.g. Ford,⁶⁸ GM,⁵⁰ and Tesla⁶⁹) have signed agreements with raw materials producers to supply their joint ventures and/or in-firm production, but may or may not directly control the exact sources of these materials, which may slightly muddy the clear delineations described in the table above. Some companies with lesser control may still choose to source batteries specifically because of the battery chemistry, but since they have no direct control, they also have less direct control over their exposure to risk from the upstream material supply chains.

- Line 39: I think it's important to mention the more local economic motivation behind these laws/regulations beyond supply chain resilience - the drive to realize the more local social and economic benefits around domestic production in place of globalization.

We appreciate the suggestion; additional language in the reworked introduction, on p. 2, copied below, has been added to recognize this fact. Additionally, further context about the history of EV development in China was added in response to another reviewer comment (see reviewer 1 comment #3).

“As a result, measurements of whole-supply chain vulnerabilities are required to better help understand economic, sustainability, and national security risks. Such measurements may help policymakers evaluate incentives and promulgate regulations on the basis of specific supply chain interventions, such as to reduce such risks, promote domestic production, or internalize economic benefits.”

3. Fig. 1: Great figure!

Thank you.

4. 137: A clear statement on what this study does to improve this detailed body of literature would be helpful. How does this study synthesize these different approaches and then improve on each area individually, or is the approach integration a novel contribution in combination with specific improvements?

We appreciate the reviewer’s suggestion to clarify the novelty of our contribution to the literature. We have done so in a largely rewritten version of the front end of the paper, highlighting in particular the novelty in terms of our focus on country exports, incorporation of the cathode material production step in the material flow analysis, and uncertainty. We addressed this comment while also reworking the introduction, context, and literature review was rewritten and consolidated in response to Reviewer #3’s specific comment #1. The new text, on p. 5, reads as follows:

“We suggest there is an important gap in the existing literature given a lack of analysis of the vulnerabilities in the global flow of multiple battery materials between countries as sources of both supply and demand across stages of the supply chain, particularly in terms of physical quantities of materials. While the criticality literature typically takes the perspective of an importing focal country, our study instead assesses global supply chain dependence on exporting focal countries. Moreover, existing studies do not explore multiple sources of uncertainty. While we do not attempt to determine potential causes of risks and vulnerabilities, nor assess the probability of such risks or vulnerabilities occurring, we aim to provide a quantitative metric to help assess vulnerabilities and evaluate potential actions to reduce vulnerabilities. In addition to specific insights about the supply chains for electric vehicle batteries, including the explicit inclusion of the cathode material manufacturing step, we provide a novel methodology incorporating material flow analysis methods with optimization methods to account for the impact of uncertainty on measurements of vulnerability. ”

5. * Fig 2: Can you include the scale in the legend? This comment goes for the following Sankeys as well. For example see Heller et al. 2020 for a US Sankey on plastics with a scale legend.

* Fig 3: Recommend moving the “b)Nickel” label to just below the dashed line, and including a dashed line to separate b) from c)

A scale in the style of Heller et al. 2020 has been added and the other graphical changes made as well; we appreciate the reference and suggestions.

6. * A lot of the results is concentrated in the MFA - which it sounded like was not the novel piece of the analysis from the Lit Review description.

We understand based on this and other comments that we may not have sufficiently described the novel contributions of our MFA approach. We hope our revision to point 4 above helps address the novelty question.

7. * 202: missing a colon in #2 after “data” // * Fig. 4: figure f) has “LFP Cathode” in the x-axis label

Fixed. The detailed notes on typos are much appreciated!

8. * Fig 4: It looks like Fig. 4 is like a condensed version of a handful of Sankey diagrams, which given the space constraints for the journal makes sense. It might be helpful to provide a full Sankey of one of these sensitivity scenarios in the Appendix to show some of It would be helpful to provide some clear context and takeaways in the discussion prior and after the figure to help the reader with the large amount of information condensed in this figure. For example, adding another sentence in the paragraph in 230 around the key takeaway, namely that if there is disruption in China, 90% of the LFP global supply chain is vulnerable to disruption (i.e., the vulnerability index result).

Thank you for this suggestion. We considered multiple strategies for displaying this information, including in a Sankey diagram and determined that because the relative amounts of lithium, nickel, cobalt, and manganese used in each battery chemistry is a relatively small proportion of all end uses for these materials, a Sankey diagram would be challenging to read and distinguish across all supply chain steps. For example, the dependence of Canada on China in Figure 4a would be challenging to display well on something like Figure 2. Instead, we have added additional language in the main text, on p. 9, and in the figure caption to help clarify and make the figures easier to interpret. We copy that text below for convenience.

Main text additions: “As an illustrative example, we present an analysis of the global disruption vulnerability index of LFP and NMC cathode production for China, the country with the most extensive involvement in battery material supply chains (as described in Fig. 1). This analysis is performed for other countries and regions in Appendix D2. Summaries for each case are presented in Figure 4, with the contribution of each step of the supply chain to the vulnerability index shown as a proportion of the end cathode material affected.”

Figure caption: “Fig. 4: Visualization of a vulnerability index for global LFP (Lithium Iron Phosphate [a-d]) and NMC (Lithium Nickel Manganese Cobalt [e-g]) cathode supply, for a lithium supply chain disruption in China.

Table 1 defines the four sensitivity cases. The amount of overall vulnerability at the cathode manufacturing step, added at each upstream step, is noted with a plus (+) sign. The solid horizontal black lines are visual aids to indicate separation of countries; semi-transparent red bars represent vulnerability that is propagated downstream. The Li supply chains differ between LFP and NMC cathodes because of differences in the distribution of countries that produce each cathode and the supply chain paths of each. Example solutions for the other NMC materials are in Appendix D1. Indirect trade and unobserved trade do not count towards the measurement of vulnerability in the optimistic case, but are counted towards the pessimistic calculation in the case that includes uncertain data. For the minimum and maximum cases, one possible solution is presented, but other distributions are possible.”

9. * Fig. 4: Why is the mining and refining of NMC Li so different than LFP? Is it just because China sources more in-country lithium for NMC?

We appreciate the very keen eye of the reviewer - the proportions of the suppliers for China's lithium should be the same in both cases, derived from the data presented in Figure 2. China and Australia were swapped in the mining step, which resulted in Australia's portion being labeled as being from China and vice versa. This error has been rectified. The refining step is accurate; since China supplies a larger percentage of global LFP cathode materials than NMC cathode materials, the supply chain pathways, and therefore the net vulnerability index, differs for the two cases. In the revision we have also added a sentence in the caption of Fig 4 to clarify this:

“The Li supply chains differ between LFP and NMC cathodes because of differences in the distribution of countries that produce each cathode and the distinct supply chain paths of each.”

10. * 364: Could you add any points on cathode recycling and recovery? Are there alternative cathode materials that are use significantly less critical materials?

We appreciate the reviewer's points that recycling, recovery, and circularity, as well as other cathode materials, are important and were not directly addressed in the original version of the paper. The additional language below, has been added to the Methodology section on p. 19, and reworked introduction on p. 2, respectively.

Methodology section: “We note that this methodology is generally flexible and can be augmented with further data as well as applied to any multi-step, multi-material global supply chain with sufficient data. For example, this study

only includes materials scrap trade codes as some measure of circularity. While current trade of end-of-life batteries and battery material supplies is minimal, if trade codes are established for such materials, or trade of such materials could be sufficiently estimated from future trade codes, this methodology could easily incorporate that additional data stream. Furthermore, we believe this methodology would be readily applicable in other supply chains, such as battery chemistries that are not yet on the market, rare earth minerals for motors, turbines, electrolyzers, solar panels, and the like, given sufficient production and trade data.”

Introduction: “Today, electric vehicle batteries mainly use lithium ion chemistries.^{3,5} The primary lithium-ion cathode chemistries are NCA (lithium nickel cobalt aluminum oxide), NMC (lithium nickel manganese cobalt oxide), and LFP (lithium iron phosphate), which depend on varying amounts of four primary ‘critical’ minerals: lithium, nickel, cobalt, and manganese, as identified in studies of mineral criticality for battery cathode materials.^{10–12} Discussion of other minerals that may be deemed as critical in other contexts, as well as the criticality of other battery chemistries under development, is in Appendix A2.”

11. * Discussion: How do you start to dive deeper into these numbers? For example, it’s clear these supply chains are very concentrated in China, and because of that distribution they are more vulnerable. Moving forward, is it possible to make this type of analysis more robust by adding how vulnerable the country is? Is it more than another country? What are the drivers (climate change, infrastructure, resources, political unrest)?

In the revision, we clarify at the end of our Introduction on p. 5, and in the Conclusions and Policy Implications section on p. 14, that we are agnostic about the potential causes of disruptions (e.g.: geopolitical strategy, economic collapse, natural disasters, political unrest, etc.), and we do not attempt to assess the probability of disruptions from different countries. Our goal is to provide a quantitative metric to help assess vulnerabilities that are expected to scale with the relative concentration of sourcing and evaluate potential actions to reduce vulnerabilities. Individual countries and administrations may have their own assessments of risks and apply their own risk preferences to make decisions. The relevant text is provided here:

Introduction: “We suggest there is an important gap in the existing literature given a lack of analysis of the vulnerabilities in the global flow of multiple battery materials between countries as sources of both supply and demand across stages of the supply chain, particularly in terms of physical quantities of materials. Moreover, existing studies do not explore multiple sources of uncertainty. While we do not attempt to determine potential causes of risks and vulnerabilities, nor assess the probability of such risks or vulnerabilities occurring, we aim to provide a quantitative metric to help assess vulnerabilities and

evaluate potential actions to reduce vulnerabilities. In addition to specific insights about the supply chains for electric vehicle batteries, including the explicit inclusion of the cathode material manufacturing step, we provide a novel methodology incorporating material flow analysis methods with optimization methods to account for the impact of uncertainty on measurements of vulnerability.”

Conclusions and Policy Implications: “Our estimates indicate a high level of potential vulnerability when considering China’s influence on the supply chain, with a vulnerability index of 92% [90% to 100%] and 80% [57% to 100%] for LFP and NMC, respectively. Though different countries and administrations may have varying thresholds of concern and risk tolerances, our methodology can help policymakers and researchers identify specific bottlenecks, key relationships, and potential levers for reduction of vulnerability or movement of production location in these interconnected supply chains, such as the key relationship between the Philippines and China regarding nickel or the dependence of South Korea and Japan on refined lithium from China. Using the diagrams, relationships, and analysis generated by this methodology, we hope to spur further research into country-specific vulnerabilities and the effects of specific disruptions on the electric vehicle battery material supply chain.”

Reviewer #3

Review of: Electric vehicle battery chemistry affects supply chain disruption vulnerabilities

Overall assessment:

This article provides an in depth study of the vulnerability of disruption in the EV battery industry due to disruption in the supply in the material systems. To this end, the authors define a disruption vulnerability index which is assessed for each material at every stage of the material manufacturing processes and use a network flow optimization model to account for the uncertainties in the data.

- A. While this article provides some quantification to the risk of disruption of some battery materials, the ranges provided are often so large that the conclusions become insignificant. In several instances, it is mentioned that the range is 0% to 100% (II. 249-250, II. 294-296, II. 326-328), which does not generate any valuable information since risk is always within this range by definition. The wide ranges presented can be expected, as it is well known that trade data are not fully harmonized and it can become difficult to assess how much a material is contained in each trade code.

We appreciate the reviewer's feedback about wide uncertainty ranges. We note that our main findings, highlighted in the abstract, have tighter ranges of uncertainty, and the fact that some materials have high uncertainty is, itself, informative about which questions we can answer with confidence and which questions leave large uncertainty, given the data available. In revising our manuscript, we have added a discussion of the uncertainty in the Results and Conclusions and Policy Recommendations sections starting on p. 13 and p.14 respectively, as follows:

Results: "Of course, a vulnerability index that ranges from 0 to 100 may not be directly useful as an exact measure of risk, but serves as an important indicator that uncertainty is large for this supply chain and thus if a policymaker is concerned about this combination of material and focal country, further investigation and detailed tracking of production and trade data may be warranted."

Conclusions and Policy Recommendations: "The uncertainty due to the *observability of trade* and *missing trade data* differs in important ways by chemistry, with Fig. 2, Fig. 3, and Fig. 5 shedding light on potential issues of data disclosure and supply chain transparency. The LFP-lithium supply chain has far less uncertainty in terms of unobservable or missing trade data than the supply chains for NMC: much of the production and trade data are observable for the entire lithium supply chain across material extraction, refining, and cathode production. In contrast, the non-lithium inputs into NMC have substantial uncertainty due to unobserved intranational flows, as well as both indirect and unobserved international trade. For example, China appears to source a

substantial amount of refined cobalt from the DRC, but this amount of trade far exceeds the amount of refined cobalt production reported in the DRC. Other studies^{23,35} indicate that “raw” cobalt undergoes initial processing within the DRC and further refining in China, as companies based in China own nearly all of the mines in the DRC.³⁷ Detailed investigation of our data indicates that nearly all of this traded refined cobalt is categorized as scrap, which also points to potential misclassification of materials as well as high uncertainty in the conversion factors used to calculate material in exports. This finding may help shed light on ongoing transparency and trade tracking issues in the cobalt supply chain, as few material flow analysis studies consider scrap trade in their assessments (see Appendix C3). Furthermore, a substantial portion of battery-grade nickel (about 20% of worldwide supply) is known to be mined and refined in Russia (or neighboring Finland) and used in the European market,^{38,39} which is not reflected in the data either. These findings suggest that further emphasis should be placed on understanding how worldwide material production and trade is tracked, particularly around refined and scrap materials. Yet this knowledge of where the data is missing could be considered a further data point on the relative vulnerability when comparing between critical minerals - policymakers may want to choose to introduce measures to minimize disruption risk for supply chains where the data is more well-understood and relative uncertainty is minimized.”

- B. A particular concern for this reviewer are assumptions underlying the trade codes used to identify the battery chemistry of the batteries being traded, which is a central issue to this paper: Trade codes are not disaggregated enough to differentiate the battery chemistry of EVs, they are merely reported based on their drive train. It is therefore unclear how the authors managed to differentiate an LFP EV from a NCM EV.

We appreciate the concern that trade codes cannot identify battery chemistries. However, our analysis does not require this information, and we have clarified the boundaries of our analysis in the revised text. Specifically, we have rewritten our introduction to include language that more clearly specifies the boundary of analysis for this study, which spans from raw material extraction to cathode production but does not include electric vehicle battery production. The revised text, starting on p. 2, reads as follows:

“The battery supply chain can be separated into three segments: upstream (mining and extraction of raw materials), midstream (processing of raw materials into battery-grade components), and downstream (cell and pack manufacturing, as well as end-of-life recycling and reuse).¹³ The supply chains for the critical minerals in these batteries differ in terms of the geography of raw material production (Fig. 1), although a few countries produce the majority of supply for each critical mineral. Arguably the most important choice is the selection of cathode material, as cathodes are over half of the cost of a battery cell and largely determine crucial battery characteristics such as energy density and

charging speed.¹⁴ While other components such as the anode (graphite)¹⁵ and electrolyte (lithium)¹⁶ may also suffer from vulnerabilities in their supply chain due to critical minerals, production data was also much more limited and thus these battery components were excluded from our scope. Most automotive manufacturers around the world are exercising increasing levels of control over their material supply chains as they design batteries for their vehicles (see Appendix A, Table A1-1). As such, this study focuses on the decision-making of firms in the upstream and midstream supply chains.”

We also have added additional language (also presented below) in our Methodology section (p. 18) to further clarify this point, specifying how the data are disaggregated across different trade codes and each supply chain step.

“We first build on the methodologies of the Material Flow Analysis literature^{16,22–26,35,51–56} to map and characterize the production and flow of materials in battery supply chains in a “data aggregation” step. To do so, we begin with the amount of material known to be produced in each of the mining, refining, and cathode production steps. Each country can produce, import, and export multiple materials at each step of the supply chain, so we convert production and trade data into quantities of ‘contained materials.’ We choose to use reported import data quantities as it tends to be more complete and accurate in comparison to reported export data.^{23,24,57,58} For example, we convert trade quantities in nickel ores and concentrates, nickel mattes, nickel sulfates, nickel waste and scrap, etc. into units of contained nickel, as prior literature^{18,28,56,59–61} indicates that some or all of these trade flows may contain nickel material that ultimately is used in battery manufacturing. See Appendix C for further discussion on the data sources and trade codes used to develop material flows, and corresponding literature. We then balance the amount of each contained material between the mining step and refining step, and between the refining step and the cathode manufacturing step, incorporating trade at each stage.”

In the revision we also include an expanded discussion of the trade codes that we use in our analysis in the supplemental information, Appendix C2 and C3.

- C. Another major concern I have is the active decision to not account for phosphorus (P) in the vulnerability assessment of batteries. P is a key material for LFP batteries and is presented as a better alternative than Ni and particularly Co, since it is not mined in the DRC. While this is true, P is considered a critical material not only as phosphorus itself, but also as elemental phosphorus (P₄) used in LFP batteries, since their supply chains are extremely concentrated and the resource may compete with the fertilizer industry used for food production (see also <https://www.nature.com/articles/s43246-022-00236-4>). Ignoring the risks linked to P in LFP batteries is neglecting a massive part of the potential risks in the production of LFP cathode materials and the conclusions that can be drawn are therefore incomplete. It is therefore the opinion of this reviewer that the comparison

presented in this paper is only partially completed and the results presented can therefore be misleading.

We appreciate the reviewer's comments on Phosphorus. In the revision, we address this concern by providing extensive context and new analysis, principally in four ways:

1. First, we now rely more extensively on the prior materials criticality literature to motivate our treatment of Phosphorus in this study. We provide a review of materials criticality in the battery materials risk analysis literature, summarizing the criteria that researchers use to determine whether or not a material is considered a critical material. An initial summary is included in Appendix A with a detailed review in Appendix B. The new text, on p. 38 and p. 43, reads:

Appendix A: "We analyzed the supply chains for four primary battery cathode materials: lithium, nickel, cobalt, and manganese. However, other materials that are sometimes considered 'critical' can be used cathode materials and/or batteries writ large, such as aluminum, phosphorus, copper, and fluorine. Studies indicate that these materials tend to be considered 'less critical' in comparison to the primary four minerals included in this study.^{10,11,70,71} We summarize various perspectives on mineral criticality in Table A2-1, with a more general discussion of mineral criticality and how to measure it in Appendix B."

Appendix B: "The literature on materials criticality assesses the risks and vulnerabilities associated with the supply of minerals and materials. These assessments often aggregate geological, technological, and economic measurements with social, regulatory, and geopolitical indicators into overarching summary metrics.⁹⁹⁻¹⁰¹ These aggregated metrics have been applied across multiple battery supply chains and battery chemistries. For example, some studies^{10,27,28} measure concentration of production with the Herfindahl-Hirschman Index and then weight them (using indices such as the World Governance Indicators, material demand, import reliance, etc.) to account for different levels of risk. While these metrics can capture a broad range of concerns, they may mask the dimensions that are important for ensuring supply availability throughout a specific product's entire supply chain.^{102,103} In this specific case, while we learn that lithium and cobalt tend to be 'more critical' than other battery minerals, we suggest that these measurements do not adequately describe the importance of specific countries and their trade inter-dependencies, especially in the context of battery material chemistry choices and their supply chains."

2. Second, we show why phosphorus is often not considered critical in the battery materials context. To do this, we compare statistics of these materials for the

battery context, including (a) the portion of global production used in batteries and (b) the portion of material demand that could be satisfied by US production, starting on p. 38:

“...The primary reason for this non- or less-critical categorization seems to be because the relative demand for such materials in batteries is (very) small relative to the overall size of the market, and the number of countries that supply the material is high. This would in theory allow for relatively easy substitution of demand from other sources given some disruption in a specific location, and the large market could allow for any slack capacity to come online to meet the gap created by the disruption. In general, if demand for the aforementioned minerals – or new minerals due to development of novel chemistries – becomes significant and/or are included on critical mineral lists, especially those related to batteries or energy technologies, then it would make sense to analyze them in a future version of this analysis.”

“We take for example the mineral phosphorus, which in Table A2-1 is considered to be critical in more contexts than the other minerals not included in the main body of the paper. Given that the criticality of phosphorus in the electric vehicle lithium ion battery context is uncertain, we also provide a Sankey diagram in Fig. A2-1. We find demand for phosphorus for LFP production was 17.5 thousand metric tons, as opposed to the roughly 30 million metric tons of phosphorus mined in 2020 (223 Mt of phosphate rock)⁷² - roughly 0.06% of total worldwide phosphorus demand. Even when narrowing down the phosphorus supply chain to phosphoric acid production, the primary precursor for the phosphorus used in LFP batteries,⁷³ this corresponds to roughly 0.075% of phosphoric acid demand. According to the USGS,¹⁷ roughly 3 million metric tons of phosphorus (23.5 million tons of phosphate rock) were mined in the United States in 2020; it is clear the United States alone could supply the world’s LFP-Phosphorus demand hundreds of times over. For further context, the USGS identifies 71 billion metric tons of phosphate rock reserves (i.e. currently reasonably economically extractable material) around the world. While we recognize it is currently non-economical to convert most of these phosphorus resources (and indeed, nickel, cobalt, and manganese, to a certain extent) to the high level of quality and purity required for LFP batteries, in the vein of the Simon-Erich wagers, we suggest that massive growth in market demand for such materials will encourage a shift in developing lower-grade resources for such applications and technological advancement to allow for such development.”

3. To provide readers with evidence to support our choice of focal materials, we include the new Table A2-1 on p. 40 in the supplemental information, summarizing which elements have been considered critical materials in a survey of recent (last 10 years) studies of multiple minerals. The studies are ordered in decreasing specificity, from electric vehicle and lithium ion batteries to batteries in general to general criticality studies.

Table A2-1. A review of the criticality of electric vehicle battery cathode materials. Y: Yes, N: No, na: not applicable

	Li	Co	Ni	Mn	Al	P	Fe	Notes
⁷⁵ Xu et al. 2022 [EV Li-Ion]	Y	Y	Y	Y	na	N*	na	Specifically responds to Spears et al.'s critique about the criticality of phosphorus. ⁶⁴ Based on their analysis, they explicitly state that "we do not believe that phosphorus is as critical a raw material from a known reserves perspective as other battery elements..."
⁷⁶ Valero et al. 2021 [EV Li-Ion]	Y	Y	Y	Y	N	N*	N	*Phosphorus is only mentioned in the fact that phosphate rock is one of the six "most common minerals throughout the 20th century and beginning of the 21st century".
²⁸ Greenwood et al. 2021 [EV Li-Ion]	Y	Y	Y	Y	na	na	na	NMC specific analysis only.
⁷⁷ Ballinger et al. 2019 [EV Li-Ion]	Y	Y	Y	N	N	N	N	Describes 3 key elements "which are significant supply risks"
¹⁸ Sun et al 2021 [Li-Ion]	Y	Y	Y	Y	na	na	na	Describes 15 commodities that include Li, Co, Ni, and Mn as the key LIB commodities that countries compete over
⁵⁶ Scott and Ireland 2020 [Li-Ion]	Y	Y	Y	na	na	na	na	Identifies four key LIB (these three plus graphite) materials based on various indicators (see Table 3).
²¹ Matos et al 2020 [Li-Ion]	Y	Y	N	Y	na	na	na	Defines these three materials as "priority raw materials"
¹¹ Wentker et al. 2019 [Li-Ion]	Y	Y	Y	Y	N	Y	N	Primarily evaluated elements of supply risk and environmental risk. Phosphorus was identified as critical because of its low substitutability and high global supply concentration, being 'more critical' than Mn but less than Li, Co, and Ni (see figures 2 and 3.)
⁵⁹ Sun et al 2019 [Li-Ion]	Y	Y	Y	Y	na	na	na	Identify these four materials as being "generally considered as the core elements for the LIBs"
¹⁰ Helbig et al 2018 [Li-Ion]	Y	Y	Y	Y	Y*	Y	Y	All are named as critical by construction; see Fig. 4 and 7. *Only Aluminum is notably 'less critical' in comparison.

	Li	Co	Ni	Mn	Al	P	Fe	Notes
¹² Olivetti et al 2017 [Li-Ion]	Y	Y	Y	Y	na	na	na	Focuses on two primary measurements: “static depletion index” and production concentration of top three countries
¹³ Granholt 2021 [Energy Storage]	Y	Y	Y	Y	na	na	na	Specifically only calls out the four as critical
⁷⁸ Hund et al 2020 [Energy Storage]	Y	Y	Y	Y	Y	na	Y	See Table 3.1 “Energy Storage”
⁷⁰ Bauer et al. 2023 [Energy]	Y	Y	Y	N	Y	N	na	See Figure 5.1. Phosphorus is identified as low supply risk and low importance to energy in the short term.
⁷¹ USGS Critical Minerals list	Y	Y	Y	Y	Y	N	na	The most general study, but this methodology is one of the standards for determining mineral criticality.

4. Finally, we have provided a Sankey diagram for phosphorus in the supplemental information with a discussion of the sources of uncertainty (as described above in our response #2 to this point C as well as below) starting on p. 39, and we mention this additional analysis in the introduction of the main text on p. 2. The Sankey diagram, specific discussion of it, and reference to this analysis are provided below for convenience.

Fig. A2-1: A Sankey diagram for global flows of phosphorus that available data suggest are involved in battery material supply chains. See Appendix C2 for further details and data sources.

“We note Fig. A2-1 displays regional production data of phosphoric acid, the primary precursor used in LFP battery cathodes, as this level of aggregation was the most detailed that was found.⁷⁴ As a result, we can only compare demand phosphorus used for LFP production to trade of phosphoric acid, and also cannot calculate a full vulnerability index in the style of the other battery materials considered in this article. Yet, even trade from international partners alone was sufficient to cover current demand for phosphorus used in LFP production – China is missing refined material when comparing trade to LFP production, but that would not affect the vulnerability index when considering China as the focal country.”

Introduction: “Today, electric vehicle batteries mainly use lithium ion chemistries.^{3,5} The primary lithium-ion cathode chemistries are NCA (lithium nickel cobalt aluminum oxide), NMC (lithium nickel manganese cobalt oxide), and LFP (lithium iron phosphate), which depend on varying amounts of four primary ‘critical’ minerals: lithium, nickel, cobalt, and manganese, as identified in studies of mineral criticality for battery cathode materials.^{10–12} Discussion of other minerals that may be deemed as critical in other contexts, as well as the criticality of other battery chemistries under development, is in Appendix A2.”

For the major reasons outlined above, I would warmly recommend to the authors to include P and P4 in their assessment, and to be more transparent about the trade codes used and their limitations in their assessment. Until this is the case, I cannot recommend that this article be published in this prestigious journal.

Thank you again for raising this important point. We hope that the addition of a phosphorus analysis (described in our response to comment C) and a transparent discussion about the trade codes and their limitations (described in our response to comment B) has fully addressed these concerns.

Specific comments

1. Structure: I find it rather confusing that you have split the introduction into background and literature review. It makes it hard to follow and you find information addressing the same topic in different places. I would encourage you to merge these sections into one logically-constructed introduction.

In the revision, we follow the suggestion and consolidate the background and literature review with the introduction.

2. II. 95-98: References?

We appreciate the reviewer’s point that this section could have used more clarity in terms of the referenced literature. With a rewriting of the introduction, we believe this has been addressed. The reworked summary paragraph, on p. 4, is included below for convenience.

“Several streams of literature have developed vulnerability-related concepts and metrics, such as literature on energy security, materials criticality, material flow analysis, input-output analysis, and supply chain disruption propagation. While a detailed literature review is provided in Appendix B, we highlight below a few key studies that have framed discussion of battery material supply chains, country-based risks, and measures of vulnerability below.”

3. LI. 106-107: What do you mean with replicate the methods? This sentence is unclear to me.

We appreciate the comment on the lack of clarity of this phrasing and have rewritten it on p. 4, copied below for convenience:

“A number of studies in the material flow analysis literature examine the structure of these supply chains in detail. Using a combination of trade data and production data at each step of the supply chain, these studies describe the relationships between geographies of production, with countries as the unit of geographic resolution. In the context of battery materials, parts of this literature focus on specific stages of the value chain, e.g. raw materials and mining, while others encompass all steps, but the scope is almost always global and limited to one specific battery material – lithium,²² cobalt,^{23,24} nickel,²⁵ manganese.²⁶ We use these methods to reveal the underlying material production and trade networks that undergird the supply chain flows in our analysis. By including the cathode material manufacturing step, we can then compare the relative demand for each material that is required to fulfill material demand for batteries, a novel addition not found in studies of individual critical mineral material flow analyses.”

4. LI. 147-149: Why did you only consider the LIB industry? The materials are used in various applications and so disruptions can be caused as much by LIB demand as by the other user industries. Also, very importantly here, which trade codes did you use? As far as I am aware, no separate trade codes exist for battery chemistries. Did you use import or export data?

In the revision we clarify in the introduction that the plausible disruptions we characterize are not necessarily caused by LIB demand. Rather, we focus on estimating the degree of LIB supply chain vulnerability to supply chain disruptions in specific countries, and we are agnostic about the potential causes of these disruptions (e.g.: geopolitical strategy, economic collapse, natural disasters, etc.). We agree that such disruptions will affect other industries as well, but our focus for this study is on the effects for electric vehicle battery production. The revised text, now on p. 5, is copied below for convenience.

“We suggest there is an important gap in the existing literature given a lack of analysis of the vulnerabilities in the global flow of multiple battery materials

between countries as sources of both supply and demand across stages of the supply chain, particularly in terms of physical quantities of materials. While the criticality literature typically takes the perspective of an importing focal country, our study instead assesses global supply chain dependence on exporting focal countries. Moreover, existing studies do not explore multiple sources of uncertainty. While we do not attempt to determine potential causes of risks and vulnerabilities, nor assess the probability of such risks or vulnerabilities occurring, we aim to provide a quantitative metric to help assess vulnerabilities and evaluate potential actions to reduce vulnerabilities. In addition to specific insights about the supply chains for electric vehicle batteries, including the explicit inclusion of the cathode material manufacturing step, we provide a novel methodology incorporating material flow analysis methods with optimization methods to account for the impact of uncertainty on measurements of vulnerability.”

As described in prior responses above, we now provide all trade codes in the supplemental information and clarify that we do not study trade codes for electric vehicle or electric vehicle battery production but, rather, focus on the supply chain up to and including cathode production. The rationale for this scope is (1) as you suggest, data availability at the battery and vehicle level are less material specific, and (2) as indicated in the introduction on p. 3 (new text copied below for convenience), the current typical firm boundary is at the choice of battery cathode chemistry, the primary difference between different lithium ion batteries.

“Arguably the most important choice is the selection of cathode material, as cathodes are over half of the cost of a battery cell and largely determine crucial battery characteristics such as energy density and charging speed.¹⁴ While other components such as the anode (graphite)¹⁵ and electrolyte (lithium)¹⁶ may also suffer from vulnerabilities in their supply chain due to critical minerals, production data was also much more limited and thus these battery components were excluded from our scope. Most automotive manufacturers around the world are exercising increasing levels of control over their material supply chains as they design batteries for their vehicles (see Appendix A, Table A1-1). As such, this study focuses on the decision-making of firms in the upstream and midstream supply chains, ending at the current typical firm decision boundary at the choice of battery cathode chemistry.”

5. LI. 151-153: What about stocks? The concentration process of lithium brines takes sometimes years and so the mining is not equal to the available Li in a given year. Also, Co is notoriously known for its opaque supply chains and black market. How can this be mass balanced if the data are not comprehensive?

For the first issue, on stocks, we note in the revision on p. 35 that stockpiling and manufacturing losses are not typically considered in material flow analyses, and that we

do include these in our Sankey diagrams by categorizing them as not battery-related refining or manufacturing production. With respect to lithium stocks and production in particular, we clarify that the USGS production data for lithium in particular is the “end” quantity of lithium produced (after concentration).

“This appendix contains two figures that describe the theoretical and modeled supply chains for electric vehicle battery materials, respectively. The primary difference is the removal of stockpiling and manufacturing losses, which are typically not explicitly considered in material flow analysis studies.^{22,24,35,51,64} Instead, we note that any stockpiling or manufacturing losses are incorporated into our uncertainty bounds when we split our refining and manufacturing into battery-related and non-battery-related refining and manufacturing, respectively. This split is identified based on our trade code aggregations as described in Appendix C3. While there may be some lag in production processes, such as lithium brines requiring up to a year to process via evaporation,⁶⁵ USGS production data is reported in terms of final production quantities (e.g. after lithium concentration), as described in Appendix C2. Using this model, we then map current electric vehicle manufacturer decision making onto the modeled supply chain. “

We have also included an updated version of our supply chain model and figure caption, which helps to re-emphasize this point:

Fig. A1-2: The simplified supply chain flows for battery materials considered in this study (in solid or dashed black lines), with manufacturer firm boundaries as described in Table A1-1. Note the steps in light gray are not modeled in this analysis; and the portions of the supply chain relating to non-battery related materials(*) would include any stockpiling of materials or manufacturing losses.

For the second issue, on opaque supply chains, we agree that the data (including data on materials stocks) is not necessarily comprehensive (a weakness of MFA studies in general), so that is why we have identified several types of uncertain data and plotted their potential effects in our supply chains in Figures 2 and 3. We hope our detailed response to overall feedback point A also helps address these concerns.

6. Section “Measuring a Vulnerability Index”: Import and export data often disagree with each other. How was this issue treated? How did you deal with double-counting?

In the revision we clarify in the Methodology section (p. 18) and Appendix C (p. 51-52) that, consistent with other material flow analysis literature, import data was used, as that tends to be more accurate and consistently reported. The data source (IntraCen’s TradeMap) attempts to resolve these discrepancies by also incorporating additional sources of information (national and regional databases) where possible. We have added additional language to our Methodology and Appendix C2-Data Sources sections to clarify these points, copied below for convenience.

Methodology: “We first build on the methodologies of the Material Flow Analysis literature^{16,22–26,35,51–56} to map and characterize the production and flow of materials in battery supply chains in a “data aggregation” step. To do so, we begin with the amount of material known to be produced in each of the mining, refining, and cathode production steps. Each country can produce, import, and export multiple materials at each step of the supply chain, so we convert production and trade statistics into quantities of ‘contained materials.’ We choose to use reported import data quantities as it tends to be more complete and accurate in comparison to reported export data.^{23,24,57,58”}

Appendix C: “Following methodologies from the material flow analysis literature,^{16,22–26,35,51–56} we trace inter-country trade from country to country by identifying relevant Harmonized System (HS) codes. These codes, internationally standardized by the World Customs Organization to 6 digits, allow for importing countries to levy tariffs and monitor compliance with regulations (e.g. rules of origin), and as such are designed to classify and cover all internationally traded items.⁵⁷ Most sources of trade data are ultimately based off of the UN Comtrade database, which aggregates reports submitted by UN member countries on their trade at a high level, which is the database used by nearly all global material flow analysis and supply chain disruption propagation studies reviewed in this article.^{16,23–26,29–32,35,51,52,54–56,59,107} Other national and regional-level trade databases, such as the US Census’s database or Eurostat, are much more detailed, but do not cover the entirety of trade around the world, and as such are less useful when considering whole supply chains. These national and regional databases could be used to supplement data gathered from international sources, which is what is done in the data source we use. It is also important to note that both export and import data exist in these databases. Nearly all

reviewed studies in the material flow analysis and supply chain disruption propagation literatures that specified a choice between the two used import data^{23,24,29,32,35} rather than export data,²⁶ as it tends to be more complete and accurate, as many countries impose various import tariffs and are concerned about what materials enter their borders,^{23,24,57,58} so we also use reported import data figures. ”

“We use TradeMap data from IntraCen⁵⁷ as it provides the UN Comtrade data while additionally supplementing it with national and regional trade where import and export data may be conflicting or missing, as some countries do not report data to UN Comtrade. It is the opinion of the authors that this incorporation of national and regional trade data positively augments the UN Comtrade database upon which most material flow analyses base their analyses on.”

7. LI. 214-216: This is a big assumption as some countries are known to be better reporters than others. How can this be justified?

We agree that this is a large assumption and have removed it to clarify our use of a more rigorous interpretation of the proportional case that is defined in most input-output and supply chain disruption propagation literature (namely, just referring to import data shares only and making no assumptions about missing or uncertain data). The improved language for this is included in response to reviewer #1's second point of detailed feedback, on p.9 and p.10 of the main text.

In the revision also we discuss the potential for and implications of uneven reporting, and we clarify the degree to which our uncertainty analysis captures uncertainty induced by this factor. We hope the language presented in response to this reviewer's overall point A addresses this.

8. Figure 4: Do the proportions of risk assessment presented here make sense? It seems like you just added the vulnerability index at each stage for the total risk. Shouldn't it be a weighted average?

We appreciate the reviewer comment on the clarity of interpretation of the vulnerability index. In the revision we clarify in the section “Measuring a Vulnerability Index” in the Results section on p. 9 that the overall vulnerability index is not a sum of the indices *from* each stage of the supply chain but, rather, an analysis of the portion of materials that pass through a given country at any stage of the supply chain relative to the *total* end product. We've also revised the caption of Figure 4, also copied below for convenience. We describe further on in this section how this is calculated with a proportional calculation, and bound uncertainty around this calculation with an optimization solution. Further details are included in the Methodology section as well as Appendix C.

Results: “Based on these flows, we can quantify measures of supply disruption vulnerability for a focal country or focal set of countries. In this analysis, we define an index of vulnerability to supply disruption from a country i as the total percentage of the end product (in this case, battery cathode material) produced with materials that are either manufactured in or traded from country i at any step of the supply chain.”

Figure Caption: “Fig. 4: Visualization of a vulnerability index for global LFP (Lithium Iron Phosphate [a-d]) and NMC (Lithium Nickel Manganese Cobalt [e-g]) cathode supply, for a lithium supply chain disruption in China.

Table 1 defines the four sensitivity cases. The amount of overall vulnerability at the cathode manufacturing step, added at each upstream step, is noted with a plus (+) sign. The solid horizontal black lines are visual aids to indicate separation of countries; semi-transparent red bars represent vulnerability that is propagated downstream. The Li supply chains differ between LFP and NMC cathodes because of differences in the distribution of countries that produce each cathode and the supply chain paths of each. Example solutions for the other NMC materials are in Appendix D1. Indirect trade and unobserved trade do not count towards the measurement of vulnerability in the optimistic case, but are counted towards the pessimistic calculation in the case that includes uncertain data. For the minimum and maximum cases, one possible solution is presented, but other distributions are possible. For all of these cases, we mask the distribution of production and trade upstream of China’s supply at each supply chain step, because it does not affect the vulnerability index and may be distributed arbitrarily in the network flow optimization results.”

9. LI. 248: Which data did you use for this? Sources?

This is the same production data that was presented in Figure 3.

10. Figure 5: It seems that the aggregation of vulnerability indices can exceed 100%. Why and how can this be the case?

As mentioned in the response to detailed point 8, we clarify in the revision that the vulnerability index is a total measurement across each material supply chain, rather than a sum of indices for each stage of the supply chain, and it cannot exceed 100%. The usefulness of this metric would be to compare across different material supply chains, rather than across stages of the same supply chain. Furthermore, we include additional language in our Results section on p. 12 to clarify how these vulnerability indices should be aggregated, if at all, copied below for convenience:

“We aim to determine an overall vulnerability index for NMC as a material, but we suggest that any combination of the vulnerability indices presented here must be done carefully in order to avoid improper comparisons across minerals and

chemistry choices. We assume a Leontief production function for multi-material systems like NMC – namely that the mineral that has the lowest amount of supply (and thus highest vulnerability index) defines the overall vulnerability of the entire material system. We then compute an overall vulnerability index for NMC using the maximum vulnerability index across the four critical materials in each of the four cases presented.”

11. LI. 312-319: This is only repeating the results. Consider either putting it in context or removing.

We appreciate the reviewer’s suggestion that this language is duplicative with the results section. In the revision, we have removed this paragraph and distributed the key information to other paragraphs on p. 14. The primary new paragraph is provided below for convenience.

“Our estimates indicate a high level of potential vulnerability when considering China’s influence on the supply chain, with a vulnerability index of 92% [90% to 100%] and 80% [57% to 100%] for LFP and NMC, respectively. Though different countries and administrations may have varying thresholds of concern and risk tolerances, our methodology can help policymakers and researchers identify specific bottlenecks, key relationships, and potential levers for reduction of vulnerability or movement of production location in these interconnected supply chains, such as the key relationship between the Philippines and China regarding nickel or the dependence of South Korea and Japan on refined lithium from China. Using the diagrams, relationships, and analysis generated by this methodology, we hope to spur further research into country-specific vulnerabilities and the effects of specific disruptions on the electric vehicle battery material supply chain.”

12. LI. 320-334: This entire paragraph is very descriptive but it is difficult to extract the relevance and insight that your points here generate. Consider making the text more targeted towards the relevance rather than the crude facts.

We appreciate the reviewer comments about the relevance and insights about the points made in this section. We have expanded this section in response to this suggestion, with the new language starting on p. 14 copied below for convenience:

“The uncertainty due to the observability of trade and missing trade data differs in important ways by chemistry, with Fig. 2 and Fig. 3 shedding light on potential issues of data disclosure and supply chain transparency. The LFP-lithium supply chain has far less uncertainty in terms of unobservable or missing trade data than the supply chains for NMC: much of the production and trade data are observable for the entire lithium supply chain across material extraction, refining, and cathode production. In contrast, the non-lithium inputs into NMC have

substantial uncertainty due to unobserved intranational flows, as well as both indirect and unobserved international trade. For example, China appears to source a substantial amount of refined cobalt from the DRC, but this amount of trade far exceeds the amount of refined cobalt production reported in the DRC. Other studies^{23,35} indicate that “raw” cobalt undergoes initial processing within the DRC and further refining in China, as companies based in China own nearly all of the mines in the DRC.³⁷ Detailed investigation of our data indicates that nearly all of this traded refined cobalt is categorized as scrap, which also points to potential misclassification of materials as well as high uncertainty in the conversion factors used to calculate material in exports. This finding may help shed light on ongoing transparency and trade tracking issues in the cobalt supply chain, as few material flow analysis studies consider scrap trade in their assessments (see Appendix C3). Furthermore, a substantial portion of battery-grade nickel (about 20% of worldwide supply) is known to be mined and refined in Russia (or neighboring Finland) and used in the European market,^{38,39} which is not reflected in the data either. These findings suggest that further emphasis should be placed on understanding how worldwide material production and trade is tracked, particularly around refined and scrap materials. Yet this knowledge of where the data is missing could be considered a further data point on the relative vulnerability when comparing between critical minerals - policymakers may want to choose to promote supply chains where the data is more well-understood and relative uncertainty is minimized.”

13. LI. 340-343: Doesn't this indicate that the trade data on cobalt is too uncertain to be useful?

We appreciate the reviewer's point made here and hope our response to overall point A addresses this specific comment.

14. LI. 349-363: This is common knowledge and general statements that don't follow from your results. How does your study build on this knowledge base?

We are thankful for the note that this portion did not adequately incorporate results from our analysis, and provide the revised text, now on p. 16, below:

“In the U.S. policy context, the Inflation Reduction Act⁷ includes several provisions that encourage firms to change their location of production. A production tax credit equal to 10% of production costs incentivizes firms to domestically manufacture electro-active materials, such as the cathode or anode. In addition, EVs can qualify for two tax credits per vehicle, based on the geography of their supply chains: (1) those that have a minimum amount of critical minerals produced or processed domestically within free trade agreement countries and (2) a minimum amount of battery components manufactured in North America. Our results further suggest that these sourcing requirements may

not avoid vulnerabilities due to specific geographies of concern along the entire supply chain (for both requirements). Furthermore, because of the compound nature of vulnerabilities across multiple supply chain stages, reduction of vulnerability at just one across these battery material supply chains is not sufficient.”

15. LI. 364-367: I don't understand this recommendation, wouldn't this just give China even more control? Neglecting P completely skews the understanding of LFP disruption risks as well. I don't think this statement is substantiated unless a thorough analysis of P is also conducted.

We appreciate the concern about increasing reliance on China. We've revised this paragraph, now on p. 16, to better explain why we suggest this recommendation, copied below for convenience. We also hope our discussion of phosphorus in response to overall point C sufficiently addresses that portion of the comment.

“A possible complementary policy action may be funding research, development, and demonstrations that aim to improve the performance of LFP batteries as well as the lithium extraction and refining processes, and thus reduce reliance on NMC and its compound material risks, as all four critical materials studied are required in order to produce NMC, creating multiple disruption paths. LFP also has an advantage given the relatively small overall physical quantity of materials that is present in the Lithium supply chain (comparing the scales in Fig. 2 and Fig. 3) and the fact that uncertainty appears to be the lowest in the Lithium supply chains, as described previously. Immense recoverable deposits of lithium are being rapidly discovered given the recent interest in the material.⁴⁶⁻⁴⁹ In particular, the U.S. has substantial potential mining capacity (the Thacker Pass mine in Nevada is the third largest individual lithium resource in the world⁴⁸), as well as existing capabilities in both mining and refining (Fig. 2). While choosing LFP may involve other technological and economic tradeoffs, a shift toward LFP may represent an opportunity to reduce disruption vulnerability, if lithium refining operations and LFP cathode production operations in particular are diversified away from the currently high concentration in China.”

Reviewers' Comments:

Reviewer #1:

Remarks to the Author:

The authors have addressed the points of clarification I raised in my previous review.

The paper is a useful addition to the literature on supply chain vulnerability and I have no further comments prior to publication.

Reviewer #2:

Remarks to the Author:

I appreciate the authors' attention to detail in address my comments. My only remaining comment is on the formatting in Fig. 3 of the main text. Fig 3a is cut off at the bottom, and it would be helpful to have a second dashed line separator between Fig 3b and Fig 3c.

Reviewer #3:

Remarks to the Author:

Thank you very much for having addressed the comments in such a thorough manner. It is clear that major efforts were made to improve the issues raised and a very well informed response was formulated. The authors have proposed a response to all of my comments and I feel that they have done a very good job at backing up their assumptions.

While I am satisfied with most of their answers, I still feel the need to push back on the issue of phosphorus, as I believe that the arguments proposed to explude P from the analysis are not sufficient to justify that decision. In addition to this major issue, I would like to raise some additional points about the uncertainty of some of the findings and their relevance.

Regarding point A:

The authors claim that the very high uncertainty range from 100% is in itself a finding that indicates which questions can be answered using the method proposed. While I generally agree with this, I believe that more than a finding that generates knowledge for the scientific community, this finding rather points to the fact that the data used in this method is not suitable to answer the research question defiend since it is too uncertain to be meaningful. Hence the sentence added to the results "Of course, a vulnerability index that ranges from 0 to 100 may not be directly useful as an exact measure of risk, but serves as an important indicator that uncertainty is large for this supply chain and thus if a policymaker is concerned about this combination of material and focal country, further investigation and detailed tracking of production and trade data may be warranted." is to my understanding meaningless, since an indicator with 100% uncertainty has no value whatsoever. Therefore, it would be my assessment that for the values with this high an uncertainty should be communicated only as a qualitative finding, more than a quantitative one, so as to provide enough information about the nuances and reasons behind this high uncertainty. The conclusions following from such indicators can only be of qualitative nature and can be an important pointer to the data gaps and needs, but cannot be used to compare the criticality of that material. This results in a different research question that is being answered, that of the issues with trade data, more than the criticality of the given material, which cannot be reliably assessed with this methodology.

Regarding point B:

The authors write: "While other components such as the anode (graphite and electrolyte (lithium) may also suffer from vulnerabilities in their supply chain due to critical minerals, production data was also much more limited and thus these battery components were excluded from our scope." Is this a good

reason to exclude these materials? I believe that just taking into consideration some materials that have good enough data to make conclusions about the overall battery supply chain that includes materials not accounted for may be misleading.

Furthermore, the authors' response indicates that the scope of the analysis spans from raw material extraction to cathode production but does not include EV battery production, but in the text they write "The battery supply chain can be separated into three segments: upstream (mining and extraction of raw materials), midstream (processing of raw materials into battery-grade components), and downstream (cell and pack manufacturing, as well as end-of-life recycling and reuse)". Isn't the cell and pack manufacturing part of the EV battery production? Please clarify and elaborate, as the production capacity of different battery chemistry may also be a major risk in creating supply chain bottlenecks.

Regarding point C:

I do not agree with this assessment even based with the author's own references. Ref 70 is a popular science article that is not a peer reviewed or scientific. Ref 71 indicates that: "The production capacity is exceeded in 2025 just upon entering the medium term. By the middle of the medium term, the production capacity for all trajectories of phosphorus far exceeds the 2020 production and production capacity levels. Since there are not any clear substitutes for phosphorus in LFP batteries for stationary storage and electric vehicles, either production for phosphorus must increase to meet energy demand or other battery technologies must be implemented to supplement the current market share of LFPs. However, since LFP batteries are currently trending toward a higher market share, an increase in production seems the more likely solution." If anything indicating that P is likely a risk factor in the future since the demand can be expected to far exceed supply already by 2025! In page 109 the report even suggests that LFP can make Ni and Co "less critical" as P acts as a substitute material for them, but in no way is it suggested that P is less critical than other metals in this study. Moreover, in ref 10 P, scores similarly for many indicators as Ni and Co, which is in accordance with criticality assessments conducted by the EU as pointed out in the first review. While it is true that the LIB industry currently accounts for only about 5% of global P consumption, this is clearly expected to change in the future as the LFP industry grows. Hence, arguing that not looking at P is not necessary because it is not very much used today is not a good reason to not include it in this prospective analysis if by all accounts the demand for P is expected to keep growing (see <https://doi.org/10.1016/j.resconrec.2023.106951> and <https://doi.org/10.1038/s43246-022-00237-3> for recent estimates).

In your response you further claim that: "This would in theory allow for relatively easy substitution of demand from other sources given some disruption in a specific location, and the large market could allow for any slack capacity to come online to meet the gap created by the disruption". This is a huge claim considering that P is an extremely geographically concentrated material and a disruption in one country could have massive repercussions to the entire P system (see food prices rise as a result of the Russian invasion of Ukraine). LFP battery manufacturing capacity cannot be shifted to NCM production from one day to the other. Therefore, a disruption to the P system would result in a disruption to the entire LIB industry.

For the reasons outlined above, I do not believe that the author's response sufficiently addresses my comments as the scope of the study has not been expanded, leaving a major part of the LIB system unaccounted for. I maintain that any conclusions drawn from this study that compares NCM chemistries to LFP are only partially informed and can therefore be misleadingly favourable to NCM, since P is not included in the analysis.

Response to Reviewer Comments

for Nature Communications

“Electric vehicle battery chemistry affects supply chain disruption vulnerabilities”

by Anthony Cheng, Erica Fuchs, Valerie Karplus and Jeremy Michalek

We thank the three reviewers for their valuable comments, which have helped us improve the manuscript. Below we respond point-by-point to each reviewer’s comments in blue text, indicating where we have made changes to the revision to address each comment.

REVIEWER COMMENTS

Reviewer #1 (Remarks to the Author):

The authors have addressed the points of clarification I raised in my previous review.

The paper is a useful addition to the literature on supply chain vulnerability and I have no further comments prior to publication.

We greatly appreciate the thorough assessment and helpful recommendations provided by this reviewer.

Reviewer #2 (Remarks to the Author):

I appreciate the authors' attention to detail in address my comments. My only remaining comment is on the formatting in Fig. 3 of the main text. Fig 3a is cut off at the bottom, and it would be helpful to have a second dashed line separator between Fig 3b and Fig 3c.

We greatly appreciate the very thoughtful attention to detail raised by this reviewer. We identified that this error is the result of a file conversion issue and have rectified it in the newest version of the paper.

Reviewer #3 (Remarks to the Author):

Thank you very much for having addressed the comments in such a thorough manner. It is clear that major efforts were made to improve the issues raised and a very well informed response was formulated. The authors have proposed a response to all of my comments and I feel that they have done a very good job at backing up their assumptions.

We greatly appreciate the comprehensive assessment and constructive recommendations of this reviewer and would like to emphasize that their suggestions have significantly improved the quality of the paper.

While I am satisfied with most of their answers, I still feel the need to push back on the issue of phosphorus, as I believe that the arguments proposed to exclude P from the analysis are not sufficient to justify that decision. In addition to this major issue, I would like to raise some additional points about the uncertainty of some of the findings and their relevance.

Thank you for raising these remaining questions, which we have made a concerted effort to address as described below.

Regarding point A:

The authors claim that the very high uncertainty range from 100% is in itself a finding that indicates which questions can be answered using the method proposed. While I generally agree with this, I believe that more than a finding that generates knowledge for the scientific community, this finding rather points to the fact that the data used in this method is not suitable to answer the research question defined since it is too uncertain to be meaningful. Hence the sentence added to the results "Of course, a vulnerability index that ranges from 0 to 100 may not be directly useful as an exact measure of risk, but serves as an important indicator that uncertainty is large for this supply chain and thus if a policymaker is concerned about this combination of material and focal country, further investigation and detailed tracking of production and trade data may be warranted." is to my understanding meaningless, since an indicator with 100% uncertainty has no value whatsoever. Therefore, it would be my assessment that for the values with this high an uncertainty should be communicated only as a qualitative finding, more than a quantitative one, so as to provide enough information about the nuances and reasons behind this high uncertainty.

The conclusions following from such indicators can only be of qualitative nature and can be an important pointer to the data gaps and needs, but cannot be used to compare the criticality of that material. This results in a different research question that is being answered, that of the issues with trade data, more than the criticality of the given material, which cannot be reliably assessed with this methodology.

Thank you for underscoring this point. We address the reviewers' concern by raising several points for consideration. First, we argue that a vulnerability index that ranges from 0 to 100, in other words, could take on any value, is not correctly interpreted as having "100% uncertainty." Instead, it correctly provides the information that based on available data and potential assignment of trade flows the index could take on any value between 0 and 100, in contrast with those with more narrow ranges for vulnerability indices estimated for other cases. This comparison provides meaningful information for readers. Second, we feel that adopting the language of a "qualitative" assessment would mislead readers to believe that we had not performed our analysis consistently across materials. We have made every effort to use precise language to describe how we have handled this issue and provided a table (described below) to provide readers with as much detailed and comparable information as possible to draw their own conclusions. In line with this approach, in our revised version, we replace "...of risk" with "...of vulnerability" in the sentence we have added to use terminology consistently and avoid

the interpretation that our vulnerability measure uniformly or comprehensively captures risk or its relationship to “criticality.”

In the revision we move Table D1-1 to the main body of the paper and reference it as Table 2, copied below, to clarify ranges of uncertainty for the vulnerability index for each of the critical materials in the LFP and NMC chemistries for the case of China. We hope this better illustrates the degree to which the range of uncertainty depends on assumptions. In many of these cases wide bounds are only observed in the extreme bounding cases (where we assume that all missing data reflects flows that pass through the focal country to bound the maximum possible vulnerability index). While it is true in these cases that we cannot categorically rule out wide bounds, analysts who are willing to accept modest assumptions (e.g.: using known trade data only) obtain tighter bounds.

Table 2. Summary of vulnerability index results for LFP and NMC battery chemistries for the case of China

	Supply Chain	Optimistic Case (Minimum)	Base Case (Proportional)	Pessimistic Case A (Known trade only)	Pessimistic Case B (Extreme bound)
LFP	Li	90%	92%	93%	100%
	Overall	90%	92%	93%	100%
NMC	Li	57%	78%	100%	100%
	Ni	57%	58%	59%	100%
	Co	57%	70%	70%	100%
	Mn	71%	80%	76%	94%
	Overall	71%	80%	100%	100%

We also indicate this in the revision by more consistently reporting uncertainty ranges in the text using the span between the optimistic case and pessimistic case A, leaving the extreme upper bound of pessimistic case B for a more extended discussion, such as in the following paragraph from p. 10:

“We bound uncertainty in the vulnerability index in Fig. 4(b-c,f-g) using network flow optimization to minimize or maximize the portion of the supply chain that could flow through China, given uncertainty about which imports for each country map to which paths. Using only the trade data that is present and accounting for the first type of uncertainty (unobserved intranational flows), we estimate minimum and maximum bounds, as for example we do not necessarily know

where the observed production of LFP in China sources lithium from. This corresponds to corresponding to an optimistic case and pessimistic case A. With our base case proportional flow estimates [optimistic estimate to pessimistic case A estimate], we estimate that 17% [0% to 33%] of refined Li imports to the US and Canada are refined in China, increasing the vulnerability index by +2 [+0 to +3] percentage points to a total of 92% [90% to 93%] of cathode material production involving China, depending on which US and Canada imports are used for LFP cathode production. However, it is important to recognize that this scenario, while pessimistic, is not a guaranteed upper bound, as we can have uncertainty due to non-observed and indirect trade that could be attributed to production in China (uncertainties 2 and 3). As a result we include a second, more extreme upper bound (Pessimistic Case B) by simultaneously assuming that all indirect trade and missing materials originate from China, resulting in the vulnerability index for LFP increasing by +7 percentage points between the two pessimistic cases to 100%, as seen in Fig. 4(d). These results suggest that LFP battery material supply chains are highly vulnerable to a disruption in China, as even in the most optimistic case the vulnerability index is over 90%. While not noted on the figure, further analysis indicates that even if all cathode production were moved out of China, an estimated 71% of LFP cathode material would use Li inputs produced or traded from China in the proportional case.”

We also included the language below on p. 8 and p. 13 to emphasize the importance of gathering better data along the supply chain to better inform our understanding of vulnerability and risk.

Page 9:

“It is also important to note that the amount of uncertainty present in these figures is relatively high, particularly for Cobalt and Nickel, as there is a significant amount of uncertainty in supply in both the raw and refined material steps. While we capture the effect of this uncertainty in our vulnerability index calculation, we note that gathering better data along the supply chain is important to continue to refine and inform our understanding of vulnerability and risk.”

Page 13:

“Fig. 5 also summarizes vulnerability indices for Russia, the Democratic Republic of the Congo (DRC), and South Africa. With proportionality assumptions, the vulnerability index is high for cobalt in the DRC and high for manganese in South Africa, but the bounding cases show substantial uncertainty – anywhere from 0% to 90% of the NMC supply chain is observed to be vulnerable to disruptions in Russia, the DRC and South Africa. The upper range can approach 100% given uncertainty of indirect trade and unobserved data: for example, the data suggest there is not much observed vulnerability to Russia, but if we were to assume that Russia supplies all of the unobserved nickel and cobalt supply in the battery materials supply chain, the vulnerability index could approach 100%. Of course,

when the optimistic and maximally pessimistic vulnerability indices range from 0 to 100, this quantitative measure cannot serve as a precise measure of vulnerability. We suggest that such a calculation indicates that uncertainty is large for this supply chain, given available data, and thus if a policymaker is concerned about this combination of material and focal country, further investigation and detailed tracking of production and trade data would be warranted.”

Finally, the third paragraph of the Conclusions and Policy Implications section (p. 15) emphasizes how the uncertainty quantified in our analysis helps identify key data needs:

“The uncertainty due to the observability of trade and missing trade data differs in important ways by chemistry, with Fig. 2, Fig. 3, and Fig. 5 shedding light on potential issues of data disclosure and supply chain transparency. The LFP-lithium supply chain has far less uncertainty in terms of unobservable or missing trade data than the supply chains for NMC: much of the production and trade data are observable for the entire lithium supply chain across material extraction, refining, and cathode production. In contrast, the non-lithium inputs into NMC have substantial uncertainty due to unobserved intranational flows, as well as both indirect and unobserved international trade. For example, China appears to source a substantial amount of refined cobalt from the DRC, but this amount of trade far exceeds the amount of refined cobalt production reported in the DRC. Other studies^{23,35} indicate that “raw” cobalt undergoes initial processing within the DRC and further refining in China, as companies based in China own nearly all of the mines in the DRC.³⁷ Detailed investigation of our data indicates that nearly all of this traded refined cobalt is categorized as scrap, which also points to potential misclassification of materials as well as high uncertainty in the conversion factors used to calculate material in exports. This finding may help shed light on ongoing transparency and trade tracking issues in the cobalt supply chain, as few material flow analysis studies consider scrap trade in their assessments (see Appendix C3). Furthermore, a substantial portion of battery-grade nickel (about 20% of worldwide supply) is known to be mined and refined in Russia (or neighboring Finland) and used in the European market,^{40,41} which is not reflected in the data either. These findings suggest that further emphasis should be placed on understanding how worldwide material production and trade is tracked, particularly around refined and scrap materials. Yet this knowledge of where the data is missing could be considered a further data point on the relative vulnerability when comparing between critical minerals - policymakers may want to choose to promote supply chains where the data is more well-understood and relative uncertainty is minimized.”

Regarding point B:

The authors write: "While other components such as the anode (graphite and electrolyte (lithium)) may also suffer from vulnerabilities in their supply chain due to critical minerals,

production data was also much more limited and thus these battery components were excluded from our scope." Is this a good reason to exclude these materials? I believe that just taking into consideration some materials that have good enough data to make conclusions about the overall battery supply chain that includes materials not accounted for may be misleading.

Thank you for encouraging us to engage more deeply with this question. In the revision we have rewritten the above sentence on page 3 to expand our rationale. All studies have scoping limitations. We aim for ours to be fully transparent so that the reader can fully understand the context of our findings and recommendations.

While other components such as the anode (graphite)¹⁵ and electrolyte (lithium)¹⁶ may also suffer from vulnerabilities in their supply chain, the choices among these components are far more limited and thus do not offer the same kind of options to reduce vulnerability by changing technology choices. Furthermore, production data were also much more limited and thus these battery components were excluded from our scope.

Furthermore, the authors' response indicates that the scope of the analysis spans from raw material extraction to cathode production but does not include EV battery production, but in the text they write "The battery supply chain can be separated into three segments: upstream (mining and extraction of raw materials), midstream (processing of raw materials into battery-grade components), and downstream (cell and pack manufacturing, as well as end-of-life recycling and reuse)". Isn't the cell and pack manufacturing part of the EV battery production? Please clarify and elaborate, as the production capacity of different battery chemistry may also be a major risk in creating supply chain bottlenecks.

The reviewer is correct that cell and pack manufacturing are part of EV battery production broadly. We chose to not include it in our scope analysis, which is limited in the supply chain up to cathode production. We chose to restrict our analysis in this way because we aim to focus on the battery chemistry decision (namely LFP versus NMC). Beyond cathode production, to the point raised in the reviewer's earlier comments, we are unable to distinguish the trade of batteries because the chemistries are not recorded in trade data and automakers tend to be tight-lipped about the chemistries they use in their vehicles; furthermore, the structure of the electric vehicle supply chain is such that downstream firms are producing battery cells and packs from materials they source from firms in the upstream and midstream parts of the supply chain.

We suggest that the production capacity of different battery chemistries (i.e. cathode materials) is distinct from the production capacity of different battery designs (cell form factors, modules, packs, etc.). While analysis of downstream battery production is interesting, previous analyses (e.g. Zhou et al. 2021 - <https://www.osti.gov/biblio/1778934>) have already provided an analysis and methodology of vulnerabilities.

The text in our manuscript that addresses this is on p.3:

“Most automotive manufacturers around the world are exercising increasing levels of control over their material supply chains as they design batteries for their vehicles (see Appendix A, Table A1-1). As such, this study focuses on the decision-making of firms in the upstream and midstream supply chains, ending at the current typical firm decision boundary, the choice of battery cathode chemistry.”

Regarding point C:

I do not agree with this assessment even based with the author's own references. Ref 70 is a popular science article that is not a peer reviewed or scientific.

We apologize for a mistake in the submission of our revision document, as the excerpt presented in the response to reviewer comments was copied earlier but was not updated once citation numbers changed. We agree Ref. 70 is a popular science article; it was used to empirically support a claim in an earlier appendix section about Tesla's agreements with raw materials producers and is not related to the additional analysis presented about Phosphorus; the correct citations should have been #71 and #72, which was what was contained in the revised version of the article. However, we have made major changes based on this reviewer's additional comments below (including removing this sentence).

Ref 71 indicates that:

"The production capacity is exceeded in 2025 just upon entering the medium term. By the middle of the medium term, the production capacity for all trajectories of phosphorus far exceeds the 2020 production and production capacity levels. Since there are not any clear substitutes for phosphorus in LFP batteries for stationary storage and electric vehicles, either production for phosphorus must increase to meet energy demand or other battery technologies must be implemented to supplement the current market share of LFPs. However, since LFP batteries are currently trending toward a higher market share, an increase in production seems the more likely solution." If anything indicating that P is likely a risk factor in the future since the demand can be expected to far exceed supply already by 2025! In page 109 the report even suggests that LFP can make Ni and Co "less critical" as P acts as a substitute material for them, but in no way is it suggested that P is less critical than other metals in this study. Moreover, in ref 10 P, scores similarly for many indicators as Ni and Co, which is in accordance with criticality assessments conducted by the EU as pointed out in the first review.

We appreciate the deep read of the references. Based on our analysis of the materials criticality literature presented in Table A2-1, we agree that P could be considered critical in some contexts, including EVs, and thus included the additional analysis in Fig. A2-1 and related discussion on page 39. With this in mind, we did not mean to imply that analysis of P (or other related materials) is not necessary at all, but rather that our methodology does not necessarily yield detailed insights for materials like P due to data issues (namely, in the material processing/refining step) and the size of the market. It is

important to note that the previous Ref. 71 (now Ref. 79) indicates in Figure 5.1 (as referenced in our Table A2-1) for both the short and medium term that Phosphorus scores a “1” on ‘importance to energy’ and a “1” and “2” respectively on ‘supply risk’, which is the lowest combined score of any material considered in the analysis. Thus, while Phosphorus could be considered critical, the relative criticality of the material is low in comparison to other minerals in that specific study.

While it is true that the LIB industry currently accounts for only about 5% of global P consumption, this is clearly expected to change in the future as the LFP industry grows.

We agree with the sentiment of this statement, but would like to push back on the numerical figures presented. While the source [prev. Ref. 71, now 79] indicates 5% of global P consumption is used in batteries, the analysis we present in our revision indicates that phosphorus used for LFP is about 17.5K metric tons, or "0.06% of worldwide phosphorus demand" and "roughly 0.075% of phosphoric acid demand" (the main precursor used for LFP). Even accounting for the use of phosphorus in lithium battery electrolytes, reviewing the table below, the DOE’s estimate would be inconsistent with other data sources if roughly 1.4 million metric tons of phosphorus (from ~11 million metric tons of phosphate rock) were being used in batteries every year, as the amount of lithium, cobalt, nickel, and manganese produced in the entire world in 2020 was approximately 82,500, 142,000, 300,000, and 19 million metric tons, respectively. Our Sankey diagram for phosphorus also shows that LFP batteries are not the primary source of demand for phosphorus:

Fig. A2-1: A Sankey diagram for global flows of phosphorus that available data suggest are involved in battery material supply chains. See Appendix C2 for further details and data sources.

To reflect the suggestions made by the reviewer in the appendix, in the revision we have now included Table A2-2 and additionally rewritten the large majority of the first two paragraphs of Appendix A2, copied below for reference:

“We analyzed the current vulnerability of the supply chains for four primary battery cathode materials: lithium, nickel, cobalt, and manganese, which are often considered as the primary ‘critical’ elements for EV batteries.^{10–12} However, other materials that are sometimes considered ‘critical’ can be used cathode materials and/or batteries writ large, such as graphite, aluminum, phosphorus, copper, and fluorine. We summarize various perspectives on mineral criticality in Table A2-1, with a more general discussion of mineral criticality and how to measure it in Appendix B. The primary reason for this non- or less-critical categorization seems to be because the relative demand for such materials in batteries is small relative to the overall size of the market, and the number of countries that supply the material is high, as seen in Table A2-2. In general, if demand for the aforementioned minerals – or new minerals due to development of novel chemistries – becomes significant and/or are included on critical mineral lists, especially those related to batteries or energy technologies, then more detailed analysis would be useful in better understanding the vulnerabilities present in their supply chains. While we focus on Lithium, Cobalt, Nickel, and Manganese due to a general understanding that these are the most critical materials at the current moment and the fact that there exists sufficient data for these materials, likely due to their status as the most critical minerals, other materials that are not yet considered as critical typically have data issues, making it challenging to apply the full analysis method we have suggested here.

... With the massive projected growth of all minerals due to the expansion of battery supply chains and the overall energy transition, we suggest that further efforts to gather better data and make such data available to researchers and analysts will be necessary to address future concerns.

Table A2-2. Comparison of Production Geography and EV-related Material Demand, 2020^{17,18}

Material	Countries specifically listed in USGS Mineral Commodity Survey (MCS) [Number of countries]	Smallest amount produced by named country in MCS	Total Global Production	Material Demand for EVs
Lithium	United States, Argentina, Australia, Brazil, Chile, China, Portugal, Zimbabwe [8]	348 metric tons (Portugal)* (*US production withheld)	82,500 metric tons	38,200 metric tons (46.3%)
Cobalt	United States, Australia, Canada, China, Dem. Rep. Congo, Cuba, Indonesia, Madagascar, Morocco, Papua New Guinea, Philippines, Russia [12]	600 metric tons (USA) (Second - Madagascar - 850 metric tons)	142,000 metric tons	87,800 metric tons (61.8%)
Nickel	United States, Australia, Brazil,	16700 metric tons (USA)	2,510,000 metric	145,800

	Canada, China, Indonesia, France (New Caledonia), Philippines, Russia [9]	(Second - Brazil - 77100 metric tons)	tons	metric tons (5.8%)
Manganese	Australia, Brazil, Burma, China, Côte d'Ivoire, Gabon, Georgia, Ghana, India, Kazakhstan, Malaysia, Mexico, South Africa, Ukraine, Vietnam [16]	121,000 metric tons (Vietnam)	18,900,000 metric tons	74,200 metric tons (0.3%)
Aluminum (Alumina)	United States, Australia, Brazil, Canada, China, Germany, Guinea, India, Indonesia, Ireland, Jamaica, Kazakhstan, Russia, Saudi Arabia, Spain, Ukraine, United Arab Emirates, Vietnam [18]	439,000 metric dry tons (Guinea)	136,000,000 metric tons	2,730 metric tons (5.2e-04% of raw Al production and 4.2e-03% of smelted Al production)
Aluminum (Bauxite)	United States, Australia, Brazil, China, Guinea, India, Indonesia, Jamaica, Kazakhstan, Russia, Saudi Arabia, Vietnam [12]	3,500,000 metric dry tons (Vietnam)	391,000,000 metric tons	
Aluminum (smelted)	United States, Australia, Bahrain, Canada, China, Iceland, India, Norway, Russia, United Arab Emirates [10]	860,000 metric tons (Iceland)	65,100,000 metric tons	
Phosphorus (Phosphate Rock)	United States, Algeria, Australia, Brazil, China, Egypt, Finland, India, Israel, Jordan, Kazakhstan, Mexico, Morocco, Peru, Russia, Saudi Arabia, Senegal, South Africa, Togo, Tunisia, Turkey, Uzbekistan, Vietnam [23]	577,000 metric tons (Mexico) ≈ 74,000 metric tons contained P	219,000,000 metric tons ≈ 28,000,000 metric tons contained P	17,500 metric tons contained P (0.06% of P demand)

Hence, arguing that not looking at P is not necessary because it is not very much used today is not a good reason to not include it in this prospective analysis if by all accounts the demand for P is expected to keep growing (see <https://doi.org/10.1016/j.resconrec.2023.106951> and <https://doi.org/10.1038/s43246-022-00237-3> for recent estimates).

We agree that looking at P in a future-looking, prospective paper would be a very interesting and worthwhile avenue of future study, and we include our analysis of the current P system in Appendix A2.

In your response you further claim that: "This would in theory allow for relatively easy substitution of demand from other sources given some disruption in a specific location, and the large market could allow for any slack capacity to come online to meet the gap created by the disruption". This is a huge claim considering that P is an extremely geographically concentrated material and a disruption in one country could have massive repercussions to the entire P system (see food prices rise as a result of the Russian invasion of Ukraine). LFP battery manufacturing capacity cannot be shifted to NCX production from one day to the other. Therefore, a disruption to the P system would result in a disruption to the entire LIB industry.

While Phosphorus is one of the more abundant elements in Earth's crust, we recognize that minerals containing it in concentrated form occur only in very few areas: known reserves are heavily concentrated in Morocco (70%), with China, Algeria, and Syria each having single-digit percentage shares [USGS 2022]. However, current production is widely spread across more than 23 countries with significant production, as shown in Table A2-2 - a broader geographical distribution than the four main minerals considered in this analysis. With increased demand, we believe phosphorus reserves continue to be identified as demand for fertilizers and other end-use products increases. For example, a recent discovery in Norway causes "experts to estimate the supply will cover EV battery, solar battery, and fertilizer needs for roughly 50 years."

<https://www.extremetech.com/science/newly-discovered-phosphate-deposit-enough-to-meet-ev-battery-solar-demand>) This is a scope and breadth similar to that of aluminum (also used in EV battery cathodes, but also not always considered critical in this context, as seen in Table A2-1).

In the revision, we remove the claim about the substitutability of phosphorus in the appendix and instead recognize the limitation of the index in accounting for the distribution of reserves. We provide the following revisions to clarify the boundaries of our index's measurements on Page 9 (Results) and Page 14 (Conclusion), respectively.

Our vulnerability index aims to estimate the potential for a disruption in one country to affect the overall supply of battery cathodes given current supply chain flows. As battery production grows and supply chains shift, vulnerability indices can be updated to reflect changing interdependencies. Our vulnerability index does not capture all possible factors that affect critical material vulnerabilities. For example, concentration of raw material reserves countries may affect the long term interdependencies in ways that are not reflected by current material flows.^{35,36}

While the index captures disruption potential in current supply chains, not necessarily in future supply chains nor in concentrations of reserves that are not currently used for production, our method could be applied to any current multi-step, multi-material global supply chain. Our results present new perspectives on uncertainty and geopolitical risk in the context of LFP, NMC, and their constituent critical minerals.

For the reasons outlined above, I do not believe that the author's response sufficiently addresses my comments as the scope of the study has not been expanded, leaving a major part of the LIB system unaccounted for. I maintain that any conclusions drawn from this study that compares NMC chemistries to LFP are only partially informed and can therefore be misleadingly favourable to NCM, since P is not included in the analysis.

In our revisions, we have made an effort to make the boundaries and applicability of our study clear, to bound potential implications of expanding the analysis, and to describe its

limitations transparently. Although our analysis is limited to the current supply chains of the four primary minerals associated with EV batteries - Li, Ni, Co, Mn - we hope our new revisions and discussion have provided additional support for our reasoning to focus on these materials. To emphasize this limitation, we have revised the abstract, copied below for convenience:

We examine the relationship between electric vehicle battery chemistry and battery material supply chain disruption vulnerability through the lens of four major battery minerals: lithium, cobalt, nickel, and manganese. We compare two leading lithium-ion electric vehicle battery chemistries – nickel manganese cobalt (NMC) and lithium iron phosphate (LFP) – by (1) defining a disruption vulnerability index characterizing the portion of each cathode’s production that uses materials extracted, refined, manufactured or traded by a given focal country and (2) using network flow optimization to bound uncertainties in the trade data. World supply is currently most vulnerable to disruptions in China for both chemistries: We estimate that 80% [71% to 100%] of NMC cathode material and 92% [90% to 93%] of LFP cathode material include any of the four minerals extracted, processed or manufactured in China. NMC has additional disruption risks due to supply chain concentrations of nickel, cobalt and manganese in other countries. We find the combined effect of country dominance across multiple supply chain stages is substantially larger than at individual steps. Our results suggest that because individual countries and trade blocs can currently dominate multiple stages of the supply chain, reducing battery supply chain risks requires understanding and simultaneously addressing vulnerabilities across multiple supply chain stages.

We also revised our text in the main body on p. 2 about additional critical materials to name phosphorus directly, as follows:

“The primary lithium-ion cathode chemistries are NCA (lithium nickel cobalt aluminum oxide), NMC (lithium nickel manganese cobalt oxide), and LFP (lithium iron phosphate), which depend on varying amounts of four primary ‘critical’ minerals: lithium, nickel, cobalt, and manganese, as identified in studies of mineral criticality for battery cathode materials.^{10–12} Discussion of other minerals that may be deemed as critical, including aluminum, phosphorus, and iron, as well as the criticality of other battery chemistries under development, is provided in Appendix A2.”

We provide an analysis of P in Appendix A2, and we note that the decision to place this analysis in the appendix, rather than in the main body, is consistent with the literature in which a small minority of the studies we reviewed (2 of 15) identified P as a critical battery material compared to a large majority for Li, Co, Ni and Mn (15 of 15 for Li and Co, 14 of 15 for Ni and 12 of 15 for Mn).

Table A2-1. A review of the criticality of electric vehicle battery cathode materials.

Y: Yes, N: No, na: not applicable

	Li	Co	Ni	Mn	Al	P	Fe	Notes
⁷⁵ Xu et al. 2022 [EV Li-Ion]	Y	Y	Y	Y	na	N	na	Specifically responds to Spears et al.'s critique about the criticality of phosphorus. ⁷³ Based on their analysis, they explicitly state that “we do not believe that phosphorus is as critical a raw material from a known reserves perspective as other battery elements...”
⁷⁶ Valero et al. 2021 [EV Li-Ion]	Y	Y	Y	Y	N	N*	N	*Phosphorus is only mentioned in the fact that phosphate rock is one of the six “most common minerals throughout the 20th century and beginning of the 21st century”.
²⁸ Greenwood et al. 2021 [EV Li-Ion]	Y	Y	Y	Y	na	na	na	NMC specific analysis only.
⁷⁷ Ballinger et al. 2019 [EV Li-Ion]	Y	Y	Y	N	N	N	N	Describes 3 key elements “which are significant supply risks”
¹⁸ Sun et al 2021 [Li-Ion]	Y	Y	Y	Y	na	na	na	Describes 15 commodities that include Li, Co, Ni, and Mn as the key LIB commodities that countries compete over
⁵⁸ Scott and Ireland 2020 [Li-Ion]	Y	Y	Y	na	na	na	na	Identifies four key LIB (these three plus graphite) materials based on various indicators (see Table 3).
²¹ Matos et al 2020 [Li-Ion]	Y	Y	N	Y	na	na	na	Defines these three materials as “priority raw materials”
¹¹ Wentker et al. 2019 [Li-Ion]	Y	Y	Y	Y	N	Y	N	Primarily evaluated elements of supply risk and environmental risk. Phosphorus was identified as critical because of its low substitutability and high global supply concentration, being more critical than Mn but less than Li, Co, and Ni (see figures 2 and 3.)
⁶¹ Sun et al 2019 [Li-Ion]	Y	Y	Y	Y	na	na	na	Identify these four materials as being “generally considered as the core elements for the LIBs”
¹⁰ Helbig et al 2018 [Li-Ion]	Y	Y	Y	Y	Y*	Y	Y	All are named as critical by construction; see Fig. 4 and 7. *Only Aluminum is notably ‘less critical’ in comparison.
¹² Olivetti et al 2017 [Li-Ion]	Y	Y	Y	Y	na	na	na	Focuses on two primary measurements: “static depletion index” and production concentration of top three countries
¹³ Granholm 2021 [Energy Storage]	Y	Y	Y	Y	na	na	na	Specifically only calls out these four as critical.
⁷⁸ Hund et al 2020 [Energy Storage]	Y	Y	Y	Y	Y	na	Y	See Table 3.1 “Energy Storage”.
⁷⁹ Bauer et al. 2023 [Energy]	Y	Y	Y	N	Y	N	na	See Figure 5.1. Phosphorus is assigned low supply risk (1-2) and low importance to energy (1) scores in both the short and medium terms, the lowest combined scores of any material analyzed with the methodology.
⁸⁰ USGS critical minerals list 2022	Y	Y	Y	Y	Y	N	na	The most general study, but this methodology is one of the standards for determining mineral criticality.

Our full revised discussion of phosphorus is copied below for convenience, excluding duplicates of the tables already shown above.

We analyzed the current vulnerability of the supply chains for four primary battery cathode materials: lithium, nickel, cobalt, and manganese, which are often considered as the primary 'critical' elements for EV batteries.¹⁰⁻¹² However, other materials that are sometimes considered 'critical' can be used cathode materials and/or batteries with large, such as graphite, aluminum, phosphorus, copper, and fluorine. We summarize various perspectives on mineral criticality in Table A2-1, with a more general discussion of mineral criticality and how to measure it in Appendix B. The primary reason for this non- or less-critical categorization seems to be because the relative demand for such materials in batteries is small relative to the overall size of the market, and the number of countries that supply the material is high, as seen in Table A2-2. With current demand for materials in electric vehicle batteries, this would in theory allow for relatively easy substitution of demand from other sources given some disruption in a specific location, and the large market could allow for any slack capacity to come online to meet the gap created by the disruption. In general, if demand for the aforementioned minerals – or new minerals due to development of novel chemistries – becomes significant and/or are included on critical mineral lists, especially those related to batteries or energy technologies, then more detailed analysis would be useful in better understanding the vulnerabilities present in their supply chains. While we focus on Lithium, Cobalt, Nickel, and Manganese due to a general understanding that these are the most critical materials at the current moment and the fact that there exists sufficient data for these materials, likely due to their status as the most critical minerals, other materials that are not yet considered as critical typically have data issues, making it challenging to apply the full analysis method we have suggested here.

We take for example the mineral phosphorus, which in Table A2-1 is considered to be critical in some contexts, like other minerals excluded from the main body of the paper. Given that the criticality of phosphorus in the electric vehicle lithium ion battery context is uncertain, we provide a Sankey diagram in Fig. A2-1, though data limitations prevent us from understanding country-level production of phosphoric acid that could be used as a precursor to LFP. We find demand for phosphorus for LFP production was 17,500 metric tons, as compared to the roughly 30 million metric tons of phosphorus mined in 2020 (223 Mt of phosphate rock)⁷³ - roughly 0.06% of total worldwide phosphorus demand.

Even when narrowing down the phosphorus supply chain to phosphoric acid production, the primary precursor for the phosphorus used in LFP batteries,⁷⁴ this corresponds to roughly 0.075% of phosphoric acid demand. According to the USGS,¹⁷ roughly 23.5 million tons of phosphate rock, or about 2.3 million metric tons of phosphorus were mined in the United States in 2020; it is clear the United States alone could supply the world's LFP-Phosphorus demand hundreds of times over. For further context, the USGS identified 71 billion metric tons of phosphate rock reserves (i.e. currently reasonably economically extractable material) around the world in 2020. While we recognize it is currently non-economical to convert most of these phosphorus resources (and indeed, nickel, cobalt, and manganese, to a certain extent) to the high level of quality and purity required for LFP batteries, in the vein of the Simon-Erich wagers, we suggest that massive growth in market demand for such materials will encourage a shift in developing lower-grade resources for such applications and technological advancement to allow for such development. With the massive projected growth of all minerals due to the expansion of battery supply chains and the overall energy transition, we suggest that further efforts to gather better data and make such data available to researchers and analysts will be necessary to address future concerns.

We note Fig. A2-1 displays regional production data of phosphoric acid, the primary precursor used in LFP battery cathodes, as this level of aggregation was the most detailed that was found.⁶⁶ As a result, we can only compare demand phosphorus used for LFP production to trade of phosphoric acid, and also cannot calculate a full vulnerability index in the style of the other battery materials considered in this article. If we were to assume all production of refined phosphorus and phosphoric in East Asia occurs in China, and only look at observed import data of those materials (rather than including domestic production), we can apply the methodology and see the results found in Table A2-3. We see that the measured vulnerability index values are not altogether that different from that of Lithium for LFP: slightly lower in the proportional case and going up to 100% in the 'known trade' pessimistic case. However, we cannot be certain of any measurements beyond the observed production of LFP in the cathode step, as we only observe inter-regional trade and thus cannot be sure if this is actually the vulnerability index for phosphorus and China.

Table A2-3. China-based Vulnerability Index Calculations for lithium and phosphorus in the LFP supply chain.

*Indicates that this number is likely an overestimate, as the middle stage of refining production only has regional data and we have assigned all production in “East Asia” to China.

	LFP - Li	LFP - P*
Optimistic Case (Minimum)	89.9%	89.9%
Base Case (Proportional)	91.6%	90.5%*
Pessimistic Case A (Known trade only)	93.2%	100%*
Pessimistic Case B (Extreme bound)	100%	100%*

Reviewers' Comments:

Reviewer #3:

Remarks to the Author:

I would like to thank the authors for their detailed answer and substantial additions to the the text. They have done a thorough qualitative analysis that provides important nuances to the issue of phosphorus. However, in my view, the comaprison of LFP and NCM chemistries remains incomplete without a full analysis of phosphorus as well, as I have expressed in previous revisions. It is my assessment that not including phosphorus in this comparison disproportionately favours LFP chemistries and may leave important issues related to its supply chain unaccounted for.

That being said, this paper does provide important insights and, within the scope of what was done, gives many policy relevant recommendations. I don't believe further revisions would be constructive at this point and fully trust the editor's judgement in taking the final decision.

Response to Reviewer Comments

for Nature Communications manuscript NCOMMS-23-29340A
“Electric vehicle battery chemistry affects supply chain disruption vulnerabilities”
by Anthony Cheng, Erica Fuchs, Valerie Karplus and Jeremy Michalek

We thank the reviewers for their valuable comments, which have helped us improve the manuscript. Below we respond point-by-point to each reviewer’s comments in blue text, indicating where we have made changes to the revision to address each comment.

REVIEWER COMMENTS

Reviewer #3 (Remarks to the Author):

I would like to thank the authors for their detailed answer and substantial additions to the text. They have done a thorough qualitative analysis that provides important nuances to the issue of phosphorus. However, in my view, the comparison of LFP and NCM chemistries remains incomplete without a full analysis of phosphorus as well, as I have expressed in previous revisions. It is my assessment that not including phosphorus in this comparison disproportionately favours LFP chemistries and may leave important issues related to its supply chain unaccounted for.

That being said, this paper does provide important insights and, within the scope of what was done, gives many policy relevant recommendations. I don't believe further revisions would be constructive at this point and fully trust the editor's judgement in taking the final decision.

We appreciate the reviewer’s candid thoughts on the validity of the work that was presented in the paper, and take the editor’s judgment to publish the article in principle as an agreement that ‘further revisions would [not] be constructive at this point’.

To ensure that the analysis presented in previous revisions has sufficient visibility within the main body of the text, we have added or revised statements about the issue of phosphorus in the main body where appropriate.

On Page 2:

Discussion of other minerals that may be deemed as critical or could become critical, including aluminum, phosphorus, and iron, as well as the criticality of other battery chemistries under development, is provided in Supplementary Text S1-2.

On Page 14:

In Supplementary Text S1-2, we discuss how phosphorus might affect this comparison if it were to be considered a critical material.

In the table legend for Table 2:

In Supplementary Text S1-2, we discuss phosphorus as a potential additional critical material for LFP.